# BAN: Detecting Backdoors Activated by Adversarial Neuron Noise

**Xiaoyun Xu**
Radboud University
xiaoyun.xu@ru.nl

**Zhuoran Liu**[*]
Radboud University
z.liu@cs.ru.nl

**Stefanos Koffas**
Delft University of Technology
s.koffas@tudelft.nl

**Shujian Yu**
Vrije Universiteit Amsterdam
s.yu3@vu.nl

**Stjepan Picek**
Radboud University
stjepan.picek@ru.nl

## Abstract

Backdoor attacks on deep learning represent a recent threat that has gained significant attention in the research community. Backdoor defenses are mainly based on backdoor inversion, which has been shown to be generic, model-agnostic, and applicable to practical threat scenarios. State-of-the-art backdoor inversion recovers a mask in the feature space to locate prominent backdoor features, where benign and backdoor features can be disentangled. However, it suffers from high computational overhead, and we also find that it overly relies on prominent backdoor features that are highly distinguishable from benign features. To tackle these shortcomings, this paper improves backdoor feature inversion for backdoor detection by incorporating extra neuron activation information. In particular, we adversarially increase the loss of backdoored models with respect to weights to activate the backdoor effect, based on which we can easily differentiate backdoored and clean models. Experimental results demonstrate our defense, BAN, is $1.37\times$ (on CIFAR-10) and $5.11\times$ (on ImageNet200) more efficient with an average 9.99% higher detect success rate than the state-of-the-art defense BTI-DBF. Our code and trained models are publicly available at `https://github.com/xiaoyunxxy/ban`.

## 1 Introduction

Deep neural networks (DNNs) are known to be vulnerable to backdoor attacks, a setting where the attacker trains a malicious DNN to perform well on normal inputs but behave inconsistently when receiving malicious inputs that are stamped with triggers [9]. The malicious model is obtained by training with poisoned data [9, 6], by tampering with the training process [22, 2], or by directly altering the model's weights [14]. Backdoors are proposed for various domains, from computer vision [9, 22, 6, 26, 40, 25] to graph data [42] and neuromorphic data [1]. Still, backdoors in computer vision received most of the attention of the research community, which also means there is a significant variety of triggers. For instance, a trigger can be a pixel patch [9], an existing image [22], dynamic perturbation [25], or image warping [26]. Various backdoor attacks target different stages of the machine learning model pipeline, inducing several threat scenarios in which defenses are developed by exploiting different knowledge.

Trigger inversion is a principled method that makes minimal assumptions about the backdoor attacks in the threat model [38, 39, 44], and it has clear advantages to training-time [18] or run-time defenses [24, 19, 11]. *Input space* trigger inversion is first introduced by Neural Cleanse (NC) [36],

---

[*] Corresponding author.

38th Conference on Neural Information Processing Systems (NeurIPS 2024).

where potential triggers for all target classes are reversed by minimizing the model loss of clean inputs. Median absolute deviations of reversed triggers from all classes are then calculated to detect triggers. More recently, input space trigger inversion methods have been shown to be ineffective against *feature space* backdoor attacks [26, 38]. To address this problem, FeatureRE [38] proposes a detection method using feature space triggers. The authors observe that features of backdoored and benign inputs are separable in the feature space by hyperplanes. Unicorn [39] formally defines and analyzes different spaces for trigger inversion problems. An invertible input space transformation function is proposed to map the input space to others, such as feature space, and to reconstruct the backdoor triggers. These trigger inversion methods can be considered as an optimization problem for the targeted class. They optimize the input images to mislead the model under various constraints and recover strong triggers in different spaces. The state-of-the-art trigger inversion method, BTI-DBF [44], increases the inversion efficiency by relaxing the dependency of trigger inversion on the target labels. BTI-DBF trains a trigger generator by minimizing the differences between benign samples and their generated poisoned version in decoupled benign features while maximizing the differences in remaining backdoor features. Feature space trigger inversion defenses are generic and effective against most backdoor attacks. However, we show that BTI-DBF may fail against BadNets, as its backdoor is not prominent in the feature space (see Section 3.4). In addition, existing works still suffer from huge computational overhead and overly rely on *prominent backdoor features*.

To resolve this shortcoming, we propose a backdoor defense called detecting **B**ackdoors activated by **A**dversarial neuron **N**oise (BAN). Our defense is inspired by the finding that backdoored models are more sensitive to adversarial noise than benign models [43, 10], and neuron noise can be adversarially manipulated to indicate backdoor activations [41]. Specifically, BAN generates adversarial neuron noise, where the weights of the victim model are adversarially perturbed to maximize the classification loss on a clean data set. Simultaneously, trigger inversion is conducted in the victim model's feature space to calculate the mask for benign and backdoor features. Clean inputs with masked feature maps are then fed to the adversarially perturbed model, based on the outputs of which backdoored models can be differentiated. Figure 1 presents a t-SNE visualization of the feature space for backdoored and clean models when they are perturbed with different levels of adversarial neuron noise. It can be observed that the backdoored model misclassifies parts of the data from all classes as the target class when the adversarial neuron noise increases, while the clean model has fewer misclassifications under the same level of noise. By leveraging the induced difference, BAN can successfully detect and mitigate backdoor attacks in both input and feature space.

We make the following contributions:

- We find a generalization shortcoming in current trigger inversion-based detection methods (FeatureRE [38], Unicorn [39], BTI-DBF [44]). In particular, feature space detections overly rely on highly distinguishable prominent backdoor features. We provide an in-depth analysis of trigger inversion-based backdoor defenses, showing that prominent backdoor features that are exploited by state-of-the-art defenses to distinguish feature space backdoors may not be suitable for the identification of input space backdoors.
- We propose detecting Backdoors activated by Adversarial neuron Noise (BAN) to mitigate this generalization shortcoming by introducing neuron noise into feature space trigger inversion. BAN includes an adversarial learning process to incorporate neuron activation information into the inversion-based backdoor detection. Experimental results demonstrate that BAN is $1.37\times$ (on CIFAR-10) and $5.11\times$ (on ImageNet200) more efficient with a 9.99% higher detect success rate than the state-of-the-art defense BTI-DBF [44].
- We also exploit the neuron noise to further design a simple yet effective defense for removing the backdoor, such that we build a workable framework.

## 2 Backdoors

**Attacks.** Backdoor attacks [9, 22, 2, 26, 25, 40, 35, 8, 20, 3] refer to injecting a secret functionality into the victim model that is activated through malicious inputs that contain the trigger. To this end, substantial research has been proposed by poisoning a small percentage of training data using small and static triggers [9, 6, 22]. Early attacks generate backdoors in input space, where BadNets [9] is the first backdoor attack in DNNs. Blend [6] proposed three injection strategies to blend translucent images into inputs of DNNs as triggers. The authors controlled the transparency of the trigger to allow the trade-off between strength of attack and invisibility. Although these attacks work well,

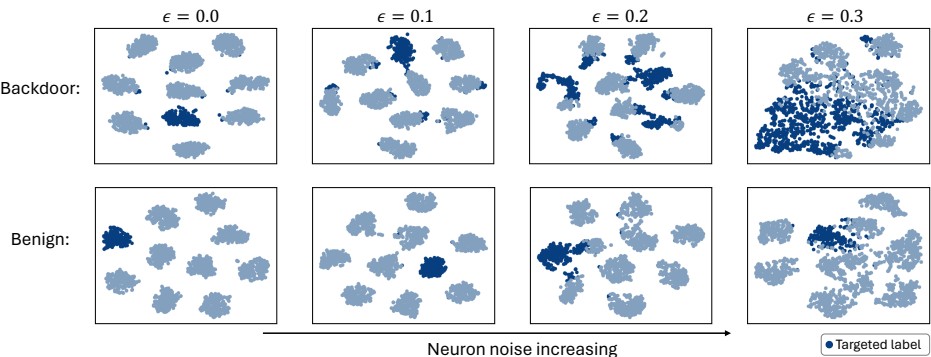

Figure 1: The feature plots of backdoor and benign models with neuron noise using ResNet18 on CIFAR-10. The darker blue represents the target label. As noise increases, the backdoor model identifies more inputs from each class as the target label. The clean model has fewer errors, and there is no significant increase in the number of misclassifications to the target class.

their triggers are still perceptible to humans and can be easily detected by backdoor defenses, such as Activation Clustering (AC) [4] and NC [36].

Dynamic and imperceptible triggers [26, 25, 40, 8], including feature space backdoor triggers [20, 25], are explored to bypass both human observers and input space defenses. IAD [25] designs input-specific triggers. To evaluate the uniqueness of dynamic triggers, the authors designed a cross-trigger test to determine whether the trigger of one input is reusable to others. WaNet [26] proposes warping-based triggers, which are unnoticeable and smooth in the input space. Bpp [40] exploits vulnerabilities in the human visual system and, by using image quantization and dithering, introduces invisible triggers that are stealthier than previous attacks.

Adaptive backdoor attacks [31, 28, 24] are built to systematically evaluate defenses, where attacks discourage the indistinguishability of latent representations of poisoned and benign inputs in the feature space. Adap-Blend [28] divides the trigger image into 16 pieces and randomly applies only 50% of the trigger during data poisoning. They use the full trigger image at inference time to mislead the poisoned model. SSDT [24] considers the combination of source-specific and dynamic-trigger backdoors. Only the inputs from victim classes (source) with the dynamic triggers are classified to the target labels, which encourages the generalization of more diverse trigger patterns from different inputs. In this paper, we evaluate BAN against various types of attacks, including input space, feature space, and adaptive attacks, to provide a systematic evaluation resembling practical threats.

**Defenses.** Trigger inversion-based backdoor defense is considered one of the most practical and general defenses against backdoors [38, 39, 44]. The recovered trigger is used to determine whether the model is backdoored. For example, NC [36] reverses input space triggers to detect backdoors by selecting significantly smaller triggers in size. Other methods, such as ABS [21] and FeatureRE [38], usually determine whether there is a backdoor based on the attack success rate of the trigger. More specifically, given a DNN model $f$ and a small set of clean data $\mathcal{D}_c = \{\mathbf{x}_n, y_n\}_{n=1}^N$, NC recovers the potential trigger by solving the following objective:

$$\min_{\mathbf{m}, \mathbf{t}} \mathcal{L}(f((1 - \mathbf{m}) \odot \mathbf{x} + \mathbf{m} \odot \mathbf{t}), y_t) + \lambda |\mathbf{m}|, \tag{1}$$

where $(\mathbf{x}, y) \in \mathcal{D}_c$ and $\mathbf{m}$ is the trigger mask, $\mathbf{t}$ is the trigger pattern, and $y_t$ is the target label. The mask $\mathbf{m}$ determines whether the pattern will replace the pixel. $\mathcal{L}$ is the cross-entropy loss function. Most prior works [12, 30, 21, 38, 39] follow this design to conduct trigger inversion for all possible target labels. For example, Tabor [12] adds more constraints to the NC optimization problem according to the overly large (large size triggers but no overlaps with the true trigger) and overlaying (overlap the true trigger but with redundant noise) problem of NC. Moving from input space to feature, FeatureRE [38] utilizes feature space constraint according to an observation that neuron activation values representing the backdoor behavior are orthogonal to others. Unicorn [39] proposes a transformation function that projects from the input space to other spaces, such as feature space [26] and numerical space [40]. Then, the authors conduct trigger inversion in different spaces.

Unlike previous NC-style methods, recent works [44, 37] explore different optimization objectives to avoid the time-consuming optimization for all possible target classes or to avoid the fixed mask-pattern design as advanced attacks utilize more complex and dynamic triggers. BTI-DBF [44] takes advantage of the prior knowledge that benign and backdoored features are decoupled in the feature space. It distinguishes them by the optimization objective, where benign features contribute to the correct predictions and backdoored features lead to wrong predictions. Based on the decoupled features, BTI-DBF trains a trigger generator by minimizing the difference between benign samples and their generated version according to the benign features and maximizing the difference according to the backdoored features. Feature space backdoor defenses are developed based on the fact that backdoor features are highly distinguishable from benign features. However, this finding is not consistently valid for input space attacks where the feature difference is small (See Sections 3.4).

## 3 BAN Method

### 3.1 The Pipeline of Training Backdoor Models

For brevity, consider an $L$-layer fully connected network $f$ (similar principles apply to convolutional networks) that has $l = l_1 + l_2 + \cdots + l_L$ neurons. $\mathbf{x}_n \in \mathbb{R}^{d_x}$ and $y_n \in \{0,1\}^{d_y}$ are the $n_{\text{th}}$ image and its label in $d_x$ and $d_y$ dimensional spaces, respectively. The attacker creates a poisoned dataset $\mathcal{D}_p$ by poisoning generators $G_X$ and $G_Y$ for a subset of the whole training dataset, i.e., $\mathcal{D}_p = \mathcal{D}_c \cup \mathcal{D}_b$. $\mathcal{D}_c$ is the original clean data. $\mathcal{D}_b$ is the poisoned backdoor data, $\mathcal{D}_b = \{(\mathbf{x}', y')|\mathbf{x}' = G_X(\mathbf{x}), y' = G_Y(y), (\mathbf{x}, y) \in \mathcal{D} - \mathcal{D}_c\}$. In all-to-one attacks, $G_Y(y) = y_t$, $y_t$ is the attacker-specified target class. In our experiments, we consider the dirty-label attack. In all-to-all attacks, ususally $G_Y(y) = (y + 1)$ [44, 26, 40], which is also what we chose in our experiments. In the training stage, the backdoor is injected into the model by training with $\mathcal{D}_p$, i.e., minimizing the training loss on $\mathcal{D}_p$ to find the optimal weights and bias $(\mathbf{w}^*, \mathbf{b}^*)$:

$$\min_{\mathbf{w}, \mathbf{b}} \mathcal{L}_{\mathcal{D}_p}(\mathbf{w}, \mathbf{b}) = \mathbb{E}_{(\mathbf{x}, y) \in \mathcal{D}_p} \ell(f(\mathbf{x}; \mathbf{w}, \mathbf{b}), y), \tag{2}$$

where $\ell(\cdot, \cdot)$ is the cross-entropy. In the inference stage, the backdoored model predicts an unseen input $\hat{\mathbf{x}}$ as its true label $\hat{y}$ but predicts $G_X(\hat{\mathbf{x}})$ as $G_Y(\hat{y})$: $f(G_X(\hat{\mathbf{x}}); \mathbf{w}, \mathbf{b}) = G_Y(\hat{y})$.

### 3.2 Threat Model

**Attacker's goal.** We consider the attacker to be the pre-trained model's provider. The attacker aims to inject stealthy backdoors into the pre-trained models. The pre-trained models perform well on clean inputs but predict the attacker-chosen target label when receiving backdoor inputs.

**Attacker knowledge.** The attacker has white-box access to the model, including training data, architectures, hyperparameters, and model weights.

**Defender's goal and knowledge.** The main goal is to detect whether a given model is backdoored and then remove the potential backdoor according to the detection results. Following [38, 39, 44], we assume the defender has white-box access to the model and holds a few local clean samples. However, the defender does not have access to the training data and has no knowledge of the backdoor trigger.

### 3.3 Detection with Neuron Noise

Based on previous findings that adversarial noise can activate backdoors [43, 41, 10], we design a two-step method for backdoor detection. First, we search for noise on neurons that can activate the potential backdoor. Then, we decouple the benign and backdoored features using a learnable mask of the latent feature (the output before the final classification layer).

**Neuron Noise.** In backdoor models, there are two types of neurons: benign and backdoor [21]. The backdoor neurons build a shortcut from $G_X(\mathbf{x})$ to $G_Y(y)$. *Neuron noise* is generated noises added on neurons to maximize model loss on clean samples in an adversarial manner [23]. The noise on benign neurons evenly misleads prediction to all classes, while the noise on backdoor neurons tends to mislead prediction to the target label due to the backdoor shortcut, as shown in Figure 1. Therefore, backdoor models with noise behave differently from benign models with noise, as there are no backdoor neurons in the benign models.

Given the $j_{\text{th}}$ neuron connecting the $i_{\text{th}}$ layer and $(i-1)_{\text{th}}$ layer, we denote its weight and bias with $\mathbf{w}_{ij}$ and $b_{ij}$, respectively. Neuron noise can be added to the neuron by multiplying its weight and bias with a small number: $(1+\delta_{ij})\mathbf{w}_{ij}$ and $(1+\xi_{ij})b_{ij}$. Then, the output of the neuron is:

$$h_{ij} = \sigma\big((1+\delta_{ij})\mathbf{w}_{ij}^{\top}\mathbf{h}_{(i-1)} + (1+\xi_{ij})b_{ij}\big), \tag{3}$$

where $\mathbf{h}_{(i-1)}$ is the output of the previous layer and $\sigma(\cdot)$ is the nonlinear function (activation). The noise on all neurons are represented by $\boldsymbol{\delta} = [\delta_{1,1}, \cdots, \delta_{l_1,1}, \cdots, \delta_{1,L}, \cdots, \delta_{l_L,L}]$ and $\boldsymbol{\xi} = [\xi_{1,1}, \cdots, \xi_{l_1,1}, \cdots, \xi_{1,L}, \cdots, \xi_{l_L,L}]$. The $\boldsymbol{\delta}$ and $\boldsymbol{\xi}$ are optimized via a maximization to increase the cross-entropy loss on the clean data:

$$\max_{\boldsymbol{\delta},\boldsymbol{\xi}\in S} \mathcal{L}_{\mathcal{D}_c}((1+\boldsymbol{\delta})\odot\mathbf{w}, (1+\boldsymbol{\xi})\odot\mathbf{b}),$$
$$S = B(\mathbf{w};\epsilon) = \{\delta\in\mathbb{R}^l | \delta \leq \epsilon\}, \tag{4}$$

where $S$ is the ball function of the radius $\epsilon$ in the $l$ dimensional space. $\boldsymbol{\delta}$ and $\boldsymbol{\xi}$ share the ball function $S$ as the maximum noise size. The maximization in Eq. (4) can be solved by PGD [23] algorithm with a random start, as PGD can better explore the entire searching space to mitigate local minima [23].

**Feature Decoupling with Mask.** The neuron noise activates the backdoor (see Figure 1) and misleads the predictions to the target label. However, the performance has high variance when searching for noise multiple times, which we conjecture is caused by random initialization. Therefore, inspired by [44], we further introduce a feature decoupling process to enhance the effect of noise on backdoored features but maintain a decreased effect on benign features. Specifically, the network $f$ is decomposed into $g = f_1 \circ \cdots \circ f_{L-1}$ and $f_L$, where $f_{L-1}$ extracts the latent features from $\mathbf{x}$, while $f_L$ is the classification layer. Then, we use a mask $\mathbf{m}$ on top of the latent feature $g(\mathbf{x})$ to decouple benign and backdoored features. The optimization objective can be written as:

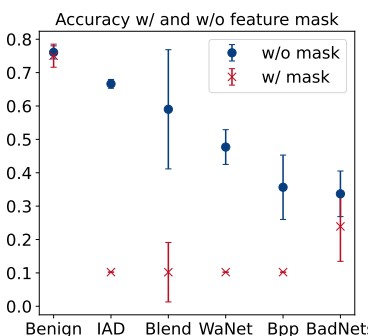
Accuracy w/ and w/o feature mask

Figure 2: Model's clean accuracy with (red dots) and without (blue dots) the mask defined in Eq. (6). Only the backdoored models are affected by the noise.

$$\min_{\mathbf{m}} \mathcal{L}(f_L\circ(g(\mathbf{x})\odot\mathbf{m}), y) - \mathcal{L}(f_L\circ(g(\mathbf{x})\odot(1-\mathbf{m})), y) + \lambda_1|\mathbf{m}|, \tag{5}$$

where $\odot$ is the element-wise product. This optimization divides the latent features into two parts through the mask while maintaining a relatively small size of $\mathbf{m}$. Note that the regularizer for $\mathbf{m}$ in Eq. (5) is necessary. Otherwise, $\mathbf{m}$ will become a dense matrix full of ones to focus on the positive part of Eq. (5) because maintaining only the positive part (without penalty of $|\mathbf{m}|$) already satisfies the optimization objective. Finally, we apply the negative mask $(1-\mathbf{m})$ on top of latent feature of $f$ to enhance the backdoor effect. The final output is:

$$f_L \circ \Big(g\big(\mathbf{x}; (1+\boldsymbol{\delta})\odot\mathbf{w}, (1+\boldsymbol{\xi})\odot\mathbf{b}\big) \odot (1-\mathbf{m})\Big). \tag{6}$$

In Figure 2, the blue dots show the accuracy after adding noise to the model. The red dots show the accuracy (with noise) while applying the feature mask to the model using Eq. (6). Our feature masks do not affect the performance of benign models. However, we see that the performance is significantly decreased for backdoored models. This decrease is caused by the model only using backdoor features (through the negative mask), which means the backdoor is activated more frequently. Finally, a suspect model is determined backdoored if the prediction using Eq. (6) has a high attack success rate. For all-to-one attacks, the misclassification will be concentrated into one label, which is the target label. For all-to-all attacks, we can do the same as for all-to-one attacks but evaluate the prediction for each class independently.

### 3.4 Improving BTI-DBF

This section shows that the most recent work, BTI-DBF [44], may fail to capture the backdoor features by its decoupling method in Table 2. We show how to patch BTI-DBF using a simple solution. The BTI-DBF is the original version, and BTI-DBF* is our improved version. The main pipeline of BTI-DBF consists of two steps: (1) decoupling benign and backdoor features and (2) trigger inversion

by minimizing the distance between benign features and maximizing the distance between backdoor features. We found that the defense's first step may introduce errors in the decoupled features. Similar to our Eq. (5), the decoupling of BTI-DBF can be written as:

$$\min_{\mathbf{m}} \mathcal{L}(f_L \circ (g(\mathbf{x}) \odot \mathbf{m}), y) - \mathcal{L}(f_L \circ (g(\mathbf{x}) \odot (1 - \mathbf{m})), y). \tag{7}$$

However, this equation has no constraints on the feature mask $\mathbf{m}$. Overall, the optimization objective is to decrease the loss. Obviously, BTI-DBF's decoupling encourages the norm of the mask to increase so that the loss will focus on the positive part and ignore the negative part because the negative part goes against the overall objective. Finally, the mask will be a dense matrix that is full of ones. The $f_L \circ (g(\mathbf{x}) \odot (1 - \mathbf{m}))$ is ignored due to multiplying by zero. We propose a simple solution to fix the problem by adding a regularizer of the size of the mask to the loss, i.e., we use our Eq. (5) as the first step of BTI-DBF*. According to Table 2, BTI-DBF* successfully overcomes BTI-DBF's shortcomings.

### 3.5 Backdoor Defense

After determining whether a suspect model is backdoored, we can fine-tune the backdoored model to remove the backdoor. However, standard fine-tuning using clean data does not effectively remove the backdoor because it does not activate it. Therefore, we propose using optimized neuron noise to fine-tune the model. In the optimization of the neuron noise, the objective is to increase the loss of $\mathcal{L}(f(\mathbf{x}), y)$. We consider both the benign and backdoor neurons to contribute to the increase in loss when optimizing the noise. Indeed, on benign neurons, the neuron noise misleads $f$ to any result other than the true label $y$, while the noise misleads $f$ to the target label $G_Y(y)$ on backdoor neurons. Therefore, a straightforward method is to decrease the loss between noise output and the true label. The loss for our noise fine-tuning can be written as:

$$\min_{\mathbf{w}, \mathbf{b}} \mathcal{L}(f(\mathbf{x}; \mathbf{w}, \mathbf{b}), y) + \lambda_2 \mathcal{L}(f(\mathbf{x}; (1 + \boldsymbol{\delta}) \odot \mathbf{w}, (1 + \boldsymbol{\xi}) \odot \mathbf{b}), y). \tag{8}$$

## 4 Experimental Results

**Datasets and Architectures.** The datasets for our experiments include CIFAR-10 [16], GTSRB [33], Tiny-ImageNet [17], and a subset of ImageNet-1K [7]. The ImageNet subset contains 200 classes, which is referred to as ImageNet200. BAN is evaluated using three architectures: ResNet18 [13], VGG16 [32], and DenseNet121 [15]. Please refer to Appendix B.1 for more details.

**Attack Baselines.** Our experiments are conducted using seven attacks: BadNets [9], Blend [6], WaNet [26], IAD [25], BppAttack [40], Adap-Blend [28], and SSDT [24], which are commonly used in other works [44, 38, 39, 46, 29]. The BadNets [9] and Blend [6] are designed for input space. WaNet [26], IAD [25], and BppAttack [40] are designed for feature space. Adap-Blend [28] and SSDT [24] (for adaptive evaluation) are state-of-the-art attacks that have been recently introduced to bypass backdoor defenses. The main idea of Adap-Blend [28] and SSDT [24] is to obscure the latent separation in features of benign and backdoor samples. More details can be found in Appendix B.2.

**Defense Baselines.** BAN is compared to five representative methods: Neural Cleanse (NC) [36], Tabor [12], FeatureRE [38], Unicorn [39], and BTI-DBF [44]. NC and Tabor are designed for input space attacks, while the other three are designed for feature space and dealing with the latest advanced attacks. BAN uses only 1% and 5% of training data for detection and defense, respectively. Note the data used for our defense is not used for training models, i.e., the defender has no knowledge of the model to be defended. More details can be found in Appendices B.3 and B.4. We use bold font to denote the best results.

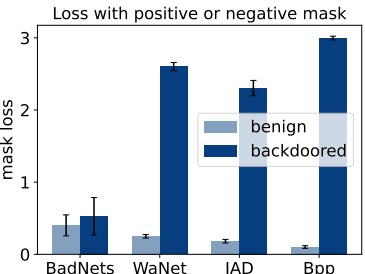

Figure 3: BadNets features are weaker when using the mask to disentangle the benign and backdoor features. Defenses that are biased towards large differences may not work in cases like BadNets.

### 4.1 The Performance of Backdoor Detection

**Main Results.** In Table 1, BAN shows better results on CIFAR-10 than all baselines, especially on advanced attacks. Results on

Table 1: The detection results under different model architectures on CIFAR-10. The "Bd." refers to the number of models the defense identifies as backdoored. The "Acc." refers to detection success accuracy. The best results are marked in bold. BTI-DBF* refers to an improved version (details in Section 3.4).

| Model | Attack | NC | | Tabor | | FeatureRE | | Unicorn | | BTI-DBF* | | Ours | |
|---|---|---|---|---|---|---|---|---|---|---|---|---|---|
| | | Bd. | Acc. | Bd. | Acc. | Bd. | Acc. | Bd. | Acc. | Bd. | Acc. | Bd. | Acc. |
| ResNet18 | No Attack | 0 | **100%** | 0 | **100%** | 2 | 90% | 6 | 70% | 0 | **100%** | 0 | **100%** |
| | BadNets | 20 | **100%** | 20 | **100%** | 14 | 70% | 18 | 90% | 18 | 90% | 20 | **100%** |
| | Blend | 20 | **100%** | 20 | **100%** | 20 | **100%** | 19 | 95% | 20 | **100%** | 18 | 90% |
| | WaNet | 11 | 55% | 8 | 40% | 15 | 75% | 20 | **100%** | 18 | 90% | 20 | **100%** |
| | IAD | 0 | 0% | 0 | 0% | 15 | 75% | 11 | 55% | 20 | **100%** | 20 | **100%** |
| | Bpp | 0 | 0% | 1 | 5% | 12 | 60% | 17 | 85 % | 20 | **100%** | 20 | **100%** |
| VGG16 | No Attack | 0 | **100%** | 0 | **100%** | 3 | 85% | 6 | 70% | 6 | 70% | 0 | **100%** |
| | BadNets | 18 | 90% | 16 | 80% | 13 | 65% | 16 | 80% | 18 | 90% | 19 | **95%** |
| | Blend | 19 | **95%** | 19 | **95%** | 16 | 80% | 18 | 90% | 16 | 80% | 17 | 85% |
| | WaNet | 10 | 50% | 9 | 45% | 12 | 60% | 18 | 90% | 16 | 80% | 20 | **100%** |
| | IAD | 0 | 0% | 0 | 0% | 8 | 40% | 17 | 85% | 20 | **100%** | 20 | **100%** |
| | Bpp | 9 | 45% | 10 | 50% | 5 | 25% | 15 | 75% | 14 | 70% | 18 | **90%** |
| DenseNet121 | No Attack | 0 | **100%** | 0 | **100%** | 5 | 75% | 8 | 60% | 3 | 85% | 0 | **100%** |
| | BadNets | 18 | 90% | 20 | **100%** | 19 | 95% | 15 | 75% | 17 | 85% | 20 | **100%** |
| | Blend | 20 | **100%** | 20 | **100%** | 12 | 60% | 18 | 90% | 19 | 95% | 20 | **100%** |
| | WaNet | 13 | 65% | 10 | 50% | 20 | **100%** | 17 | 85% | 14 | 70% | 19 | 95% |
| | IAD | 0 | 0% | 0 | 0% | 14 | 70% | 16 | 80% | 14 | 70% | 19 | **95%** |
| | Bpp | 0 | 0% | 0 | 0% | 16 | 80% | 8 | 40% | 16 | 80% | 20 | **100%** |
| Average | | 60.56% | | 59.17% | | 72.5% | | 78.61% | | 86.39% | | **97.22%** | |

other datasets are presented in Appendix C. We note that the advanced detection methods (FeatureRE, Unicorn, and BTI-DBF) perform worse than NC on simple attacks (BadNets and Blend). We hypothesize this is because the backdoor features generated by BadNets are less obvious on feature channels than advanced attacks in the feature space, such as WaNet and IAD. Figure 3 shows that BadNets features are weaker than features from advanced attacks using the feature mask in Eq. (5). Specifically, we optimize the feature mask to disentangle the benign and backdoor features for four attacks. Then, we compute two cross-entropy loss values using the positive mask ($\mathbf{m}$) and the negative mask ($1 - \mathbf{m}$) for benign and backdoor features, respectively. The average loss values and standard deviation for four models for each attack are plotted in Figure 3. The negative loss value of BadNets is much smaller than others, which means BadNets features are weaker than others with regard to misleading the model to the backdoor target. Recent defenses usually add more regularizers to their losses and optimization objectives to counteract powerful backdoor attacks. These regularizers encourage the capturing of strong features but omit weak ones. Thus, recent advanced detections can perform worse on BadNets than NC.

**Time consumption.** BAN is efficient and scalable as we do not iterate over all target classes. Figure 4 demonstrates that BAN uses substantially less time than all baselines. BAN is also more scalable for larger architectures or datasets. BIT-DBF (76.90s) is 1.37× slower than BAN (55.95s) on CIFAR-10 with ResNet18, and BIT-DBF (5,792.51s) without pre-training is 5.11× slower than ours (1,132.85s) on ImageNet200 with ResNet18. FeatureRE (297.74s) is 5.32× slower than ours on CIFAR-10 with ResNet18, and it (6,053.62s) is 45.14× slower on CIFAR-10 with DenseNet121 than ours (134.10s).

## 4.2 The Performance of Backdoor Defense

A complete backdoor defense framework should include both detection and defense. The goal of defense is to decrease the attack success rate (ASR) of backdoor triggers. Detection before the defense is also necessary because the defense usually decreases the performance on benign inputs [47, 44, 38, 46, 48]. In Section 3.5, we propose a simple and effective fine-tuning method using the noise that activates the backdoor. Table 3 compares BAN with three baselines: plain fine-tuning, FeatureRE [38], and BTI-DBF(U) [44]. Plain fine-tuning refers to training the backdoor model using the same hyperparameters as BAN but without the noise loss in Eq. (8). The FeatureRE [38]

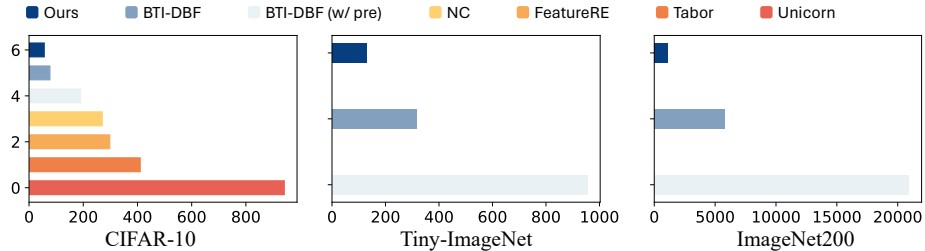

Figure 4: Time consumption of detection baselines on ResNet18 (in seconds) for all three datasets. BAN uses significantly less time than the baselines.

Table 2: The detection results of BTI-DBF and BTI-DBF* using ResNet18 and CIFAR-10.

| Attack | BTI-DBF | | BTI-DBF* | | Ours | |
|---|---|---|---|---|---|---|
| | Bd. | Acc. | Bd. | Acc. | Bd. | Acc. |
| No Attack | 0 | **100%** | 0 | **100%** | 0 | **100%** |
| BadNets | 5 | 25% | 18 | 90% | 20 | **100%** |
| Blend | 0 | 0% | 20 | **100%** | 18 | 90% |
| WaNet | 7 | 35% | 18 | 90% | 20 | **100%** |
| IAD | 19 | 95% | 20 | **100%** | 20 | **100%** |
| Bpp | 20 | **100%** | 20 | **100%** | 20 | **100%** |

and BTI-DBF(U) [44] refer to the defense methods from the respective paper. We use default hyperparameters for FeatureRE [38] and BTI-DBF(U) [44]. For plain fine-tuning and BAN, we use a small learning rate (0.005) to avoid jumping out of the current optimal parameters of the well-trained model. Then, we use $\lambda_2$ (0.5) for Eq. (8) for the trade-off between robustness against backdoor and clean accuracy. We fine-tune for a short schedule of 25 epochs, as the model is well-trained. Table 8 in Appendix B.4 shows defense performance with standard deviation on different hyperparameters, which supports our choice. Tables 3 and 4 demonstrate that our fine-tuning method effectively removes the backdoor while preserving high accuracy in benign inputs. We provide a comparison between our find-tuning with ANP [41] in Table 12, Appendix C.4, as ANP uses the neuron noise for pruning backdoor neurons.

### 4.3 Defense against All-To-All Attacks

Previous works [38, 39] usually only consider all-to-one attacks, which limits the application in practical situations. In this section, we evaluate our fine-tuning method under three all-to-all attacks: WaNet-All [26], IAD-All [25], and Bpp-All [40]. FeatureRE is designed for all-to-one attacks, so we use target label 0 here for FeatureRE. FeatureRE is included to show that the all-to-one defense does not work in the all-to-all setting. BAN is capable of handling all-to-all attacks because we directly explore the neurons themselves instead of optimizing for the potential targeted label. Table 5 demonstrates the effectiveness of our method.

Table 3: Defense against 5 attacks using ResNet18. BA refers to benign accuracy on clean data.

| Attack | No defense | | Fine-tuning | | FeatureRE | | BTI-DBF(U) | | Ours | |
|---|---|---|---|---|---|---|---|---|---|---|
| | BA | ASR | BA | ASR | BA | ASR | BA | ASR | BA | ASR |
| BadNets | 93.37 | 99.41 | 92.93 | 87.81 | **93.15** | 99.79 | 91.26 | 13.12 | 92.06 | **1.97** |
| Blend | 94.60 | 100.00 | 93.07 | 99.99 | **93.20** | 39.28 | 91.86 | 100.00 | 92.72 | **4.10** |
| WaNet | 93.57 | 99.37 | 93.05 | 1.10 | **93.67** | **0.03** | 90.30 | 4.89 | 92.05 | 0.91 |
| IAD | 93.17 | 97.88 | **94.11** | 0.46 | 92.73 | **0.0** | 89.54 | 1.59 | 92.78 | 1.48 |
| Bpp | 94.29 | 99.93 | 93.85 | 4.46 | **94.21** | 98.13 | 90.61 | 2.73 | 92.54 | **2.58** |
| Average | 93.80 | 99.32 | **93.40** | 38.76 | 93.39 | 47.45 | 90.71 | 24.47 | 92.43 | **2.21** |

Table 4: Defense of backdoor attacks on Tiny-ImageNet and ImageNet200 using ResNet18.

| Dataset | Attack | No defense | | Fine-tuning | | BTI-DBF(U) | | Ours | |
|---|---|---|---|---|---|---|---|---|---|
| | | BA | ASR | BA | ASR | BA | ASR | BA | ASR |
| Tiny-ImageNet | WaNet | 58.32 | 99.85 | **51.53** | 1.3 | 39.49 | 0.96 | 50.69 | **0.86** |
| | IAD | 58.54 | 99.32 | **51.86** | 1.72 | 38.79 | **0.60** | 50.04 | 0.76 |
| | Bpp | 60.63 | 99.89 | **57.72** | 0.15 | 46.84 | 0.40 | 57.66 | **0.10** |
| ImageNet200 | WaNet | 77.01 | 99.74 | 66.71 | 0.78 | 63.47 | 1.0 | **69.95** | **0.58** |
| | IAD | 76.72 | 99.75 | 69.91 | **0.42** | 64.33 | 1.24 | **72.18** | 1.30 |
| | Bpp | 78.56 | 99.88 | 70.89 | **0.82** | 67.02 | 3.10 | **72.59** | 2.68 |

Table 5: Defense against all-to-all backdoor attacks. BA refers to benign accuracy on clean data.

| Attack | No defense | | Fine-tuning | | FeatureRE | | BTI-DBF(U) | | Ours | |
|---|---|---|---|---|---|---|---|---|---|
| | BA | ASR | BA | ASR | BA | ASR | BA | ASR | BA | ASR |
| WaNet-All | 93.60 | 91.86 | 92.65 | 18.03 | **93.33** | 91.96 | 91.30 | 1.72 | 92.29 | **1.11** |
| IAD-All | 92.96 | 90.62 | **93.19** | 3.72 | 93.06 | 91.20 | 91.36 | 3.72 | 92.31 | **1.13** |
| Bpp-All | 94.45 | 84.68 | **93.90** | 1.58 | 94.32 | 83.87 | 90.16 | 2.05 | 93.23 | **1.38** |
| Average | 93.67 | 89.05 | 93.25 | 7.78 | **93.57** | 89.01 | 90.94 | 2.49 | 92.61 | **1.21** |

## 4.4 Evaluation under Adaptive Attack

Attackers may design a specific attack for defense when they know its details [34]. In this section, we evaluate BAN against two attacks that attempt to bypass the difference between backdoor and benign features: Adap-Blend [28] and SSDT attack [24]. Both Adap-Blend [28] and SSDT attack [24] try to obscure the difference between benign and backdoor latent features. Adap-Blend [28] achieves it by randomly applying 50% of the trigger to poison the training data, while SSDT attack [24] utilizes source-specific and dynamic-triggers to reduce the impact of triggers on samples from non-victim classes. The source-specific attack refers to backdoor triggers that mislead the model to the target class only when applied to victim class samples. Table 6 demonstrates our approach is resistant to these two attacks, while other methods fail. The reason is that the backdoor features of SSDT are close to benign features in the feature space. It is difficult for other methods to distinguish between backdoor and benign features created by SSDT. Our detection method directly analyzes the model itself using neuron noise, which captures the difference between backdoor and benign models concerning parameters. See Appendix C.3 for the mitigation results of our method against these adaptive attacks.

Table 6: The detection results under adaptive attacks on using CIFAR-10 and ResNet18.

| Attack | NC | | Tabor | | FeatureRE | | Unicorn | | BTI-DBF* | | Ours | |
|---|---|---|---|---|---|---|---|---|---|---|---|---|
| | Bd. | Acc. | Bd. | Acc. | Bd. | Acc. | Bd. | Acc. | Bd. | Acc. | Bd. | Acc. |
| Adap-Patch | 18 | 90% | 15 | 75% | 17 | 85% | 20 | **100%** | 20 | **100%** | 20 | **100%** |
| SSDT | 0 | 0% | 0 | 0% | 0 | 0% | 0 | 0% | 20 | **100%** | 20 | **100%** |

## 4.5 Analysis on Prominent Features

We provide additional analysis of the phenomenon that backdoor features are more prominent for advanced attacks (WaNet, IAD, and Bpp) than weaker attacks (BadNets, Blend). Table 7 demonstrates that previous decoupling methods cannot easily pick up backdoor features from weak attacks, such as BadNets. In particular, when detecting without the $L_1$ regularizer (i.e., w/o norm), the negative feature loss of BadNets is high with a very large $L_1$ mask norm, while the Bpp has an even higher negative loss with a much smaller mask norm. The high negative loss of BadNets is actually from the sparse feature mask rather than backdoor features, i.e., there are too many zeros in $(1 - \mathbf{m})$. This indicates that BadNet backdoor features are less prominent than Bpp features, making it more challenging to decouple BadNets features.

Table 7: Loss values when feeding the benign or backdoor features into the final classification layer. The mask is optimized using Eq. (5) (w/ norm) or Eq. (7) (w/o norm). The shape of the feature mask is $512 \times 4 \times 4$, so the maximal $L_1$ norm is 8192.

| Attack | BA | ASR | $L_1$ mask norm | Positive feature loss | Negative feature loss |
|---|---|---|---|---|---|
| BadNets (w/ norm) | 93.47 | 99.71 | 2258.90 | 0.21 | 0.26 |
| BadNets (w/o norm) | 93.47 | 99.71 | 8054.45 | 0.14 | 2.17 |
| Blend (w/ norm) | 94.60 | 100.00 | 2084.62 | 0.15 | 0.20 |
| Blend (w/o norm) | 94.60 | 100.00 | 8117.90 | 0.04 | 2.22 |
| WaNet (w/ norm) | 93.88 | 99.63 | 7400.97 | 0.06 | 2.34 |
| WaNet (w/o norm) | 93.88 | 99.63 | 7702.56 | 0.05 | 2.39 |
| IAD (w/ norm) | 93.82 | 99.64 | 7898.91 | 0.03 | 2.25 |
| IAD (w/o norm) | 93.82 | 99.64 | 7895.16 | 0.03 | 2.25 |
| Bpp (w/ norm) | 94.56 | 99.97 | 7147.68 | 0.09 | 2.80 |
| Bpp (w/o norm) | 94.56 | 99.97 | 7260.31 | 0.09 | 2.78 |

# 5 Conclusions and Future Work

This paper proposes an effective yet efficient backdoor defense, BAN, that utilizes the adversarial neuron noise and the mask in the feature space. BAN is motivated by the observation that traditional defenses outperformed the latest feature space defenses on input space backdoor attacks. To this end, we provide an in-depth analysis showing that feature space defenses are overly dependent on prominent backdoor features. Experimental studies demonstrate BAN's effectiveness and efficiency against various types of backdoor attacks. We also show BAN's resistance to potential adaptive attacks. Future studies could explore a more practical detection method without assuming access to local benign samples and better strategies for decoupling features because a fixed mask in the feature is not always aligned with the benign and backdoor features.

## Acknowledgments and Disclosure of Funding

This work was partly funded by the Dutch Research Council (NWO) through the PROACT project (NWA.1215.18.014) and the CiCS project of the research programme Gravitation (024.006.037).

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

# A  Discussion

**Limitations.** Similar to existing works [36, 21, 12, 38, 39, 44], BAN assumes a local small and benign dataset. This scenario is common, considering that benign samples are available online, and the model provider may provide some samples to verify the model's performance. In addition, our fine-tuning with neuron noise may slightly decrease the performance in benign inputs, similar to other defense methods [38, 44, 47, 41, 18, 36, 48]. However, BAN provides better performance and requires less time consumption.

**Broader Impacts.** This paper proposes a complete defense that includes detecting and removing DNN backdoors. We believe our approach has a positive impact on the security of DNNs. A possible negative impact is overconfidence in robustness against backdoor attacks, as there is no theoretical guarantee always to remove the backdoor.

# B  Additional Details for Experimental Settings

## B.1  Datasets

**CIFAR-10.** The CIFAR-10 [16] contains 50,000 training images and 10,000 testing images with the size of $3 \times 32 \times 32$ in 10 classes.

**GTSRB.** The GTSRB [33] contains 39,209 training images and 12,630 testing images in 43 classes. In our experiments, the images are resized to $3 \times 32 \times 32$.

**Tiny-ImageNet.** Tiny-ImageNet [17] contains 100,000 training images and 10,000 testing images with the size $3 \times 64 \times 64$ in 200 classes.

**ImageNet.** ImageNet [7] contains over 1.2 million high-resolution images in 1,000 classes. Our experiments use a subset of the full ImageNet dataset, i.e., 200 randomly selected classes. Each class has 1300 training images and 50 testing images. The ImageNet images are scaled to $3 \times 224 \times 224$.

## B.2  Backdoor Models

BAN and defense baselines are evaluated using seven well-known attacks: BadNets [9], Blend [6], WaNet [26], IAD [25], BppAttack [40], Adap-Blend [28], and SSDT [24].

For all backdoored models, we use SGD with a momentum of 0.9, weight decay of $5 \times 10^{-4}$, and a learning rate of 0.01. We train for 200 epochs, and the learning rate is divided by 10 at the 100-th and 150-th epochs. Same to [41], we use 90% of training data for training backdoor models and 10% for the validation set.

**BadNets.** We use a $3 \times 3$ pattern to build the backdoor trigger. The poisoning rate is 5%.

**Blend.** We use the random Gaussian noise and blend ratio 0.2 for backdoor training. The poisoning rate is 5%.

**Adap-Blend.** We use the "hellokitty_32.png" and blend ratio of 0.2 to build triggers. The poisoning rate is 5%.

**SSDT.** SSDT is a Source-Specific attack, which only misleads the victim classes to the targeted class. We use class 1 as the victim and class 0 as the target class.

For other attacks and hyperparameters not mentioned, we use the default settings from the papers or their official open-source implementations.

## B.3  Defense Baselines

BAN is compared with five representative methods, including Neural Cleanse (NC) [36], Tabor [12], FeatureRE [38], Unicorn [39], and BTI-DBF [44]. In this section, we describe the hyperparameters of these defenses.

**NC and Tabor.** We use the implementation and default hyperparameters from TrojanZoo [27] for NC and Tabor. 1% of the training set is used to conduct 100 epochs of the trigger inversion. The learning rate is 0.01.

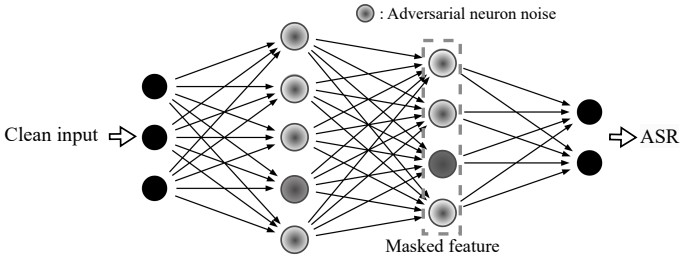

Figure 5: Illustrative diagram of BAN

**FeatureRE.** We first tried the default hyperparameters from the FeatureRE paper, but they could not work even for BadNets. We assume this is because the constraints, including feature mask size and similarity between original images and trigger images, are limited to a very small size. Therefore, we relaxed the constraints on masks and norms to find stronger triggers, including loss std bound=1.2, p loss bound=0.2, loss std bound=1.2, mask size=0.06, and learning rate=0.001. We use 1% of the training set and run FeatureRE for 400 epochs for each class.

**Unicorn.** The default hyperparameters for Unicorn are too powerful in our case, and the recovered triggers can mislead any model to the target label when applied to inputs. Every model, including benign models, is thought of as backdoored. Therefore, we slightly increase thread values of the constraints on Unicorn optimization to find proper triggers, including loss std bound=0.5, SSIM loss bound=0.1, and mask size=0.01. We use 1% of the training set and run Unicorn 40 epochs for each class.

**BTI-DBF.** We follow the default settings in the BTI-DBF [44] paper. Note that BTI-DBF uses 5% of training samples for defenses.

## B.4 BAN Settings

**Detection.** We use epsilon ($\epsilon$) to limit the noise added to neurons. Otherwise, the model weights are destroyed by the noise. The $\epsilon$ values are 0.3, 0.3, 0.2, and 0.1 for CIFAR-10, GTSRB, Tiny-ImageNet, and ImageNet200, respectively. $\epsilon$ values are decided according to the size of the images. We use smaller $\epsilon$ for larger datasets. We use the 30-step PGD algorithm to solve the optimization in Eq. (4) to find the noise, i.e., $\delta$ and $\xi$. We use SGD and the learning rate of $\epsilon/30$ for the PGD optimization. Then, we use Adam and the learning rate of 0.01 to search for 20 epochs for the feature mask using Eq. (5). The $\lambda_1$ for Eq. (5) equals 0.75. The two optimizations above use 1% of the training set. Note that this 1% of data is not used to train the backdoor models. The elements in the mask are clamped into continuous values between 0 and 1. In Figure 5, we present an illustrative diagram of BAN.

**Defense.** Our defense uses Eq. (8) to remove the backdoor. Due to limited access to benign samples and time consumption, we only use 5% of training data and 25 epochs for our fine-tuning. The trade-off hyperparameter ($\lambda_2$) is 0.5. Then, we use the most commonly used hyperparameters for the optimizer. We use SGD with momentum=0.9, weight decay=5e-4, and learning rate=0.005 as the optimizer. The plain fine-tuning uses the same hyperparameters as BAN.

Table 8 shows the ablation results on fine-tuning epochs, $\lambda_2$, and learning rate. Considering both performances on benign accuracy and removing the backdoor, our hyperparameters (LR=0.005, $\lambda_2$=0.5 and epoch=25) show the best results.

**Hardware.** All experiments are run on a single machine with 4 RTX A6000 (48GB) and 4 RTX A5000 (24GB) GPUs, CUDA 12.0.

Table 8: Ablation of hyperparameters for our defense. Each entry is the average of 5 runs.

| Attack | LR | Epoch | $\lambda_2$ | Ours | |
| | | | | BA | ASR |
|---|---|---|---|---|---|
| | 0.005 | 25 | 0.2 | $93.26 \pm 0.14$ | $2.53 \pm 0.93$ |
| | 0.005 | 25 | 0.5 | $92.64 \pm 0.19$ | $2.28 \pm 0.72$ |
| | 0.005 | 25 | 0.8 | $92.19 \pm 0.13$ | $2.89 \pm 0.94$ |
| | 0.005 | 25 | 1.0 | $92.04 \pm 0.29$ | $2.74 \pm 0.49$ |
| | 0.005 | 10 | 0.5 | $92.97 \pm 0.29$ | $2.62 \pm 0.44$ |
| Bpp | 0.005 | 50 | 0.5 | $92.57 \pm 0.09$ | $2.80 \pm 0.73$ |
| | 0.005 | 75 | 0.5 | $92.51 \pm 0.16$ | $2.34 \pm 0.17$ |
| | 0.001 | 25 | 0.5 | $93.57 \pm 0.17$ | $2.89 \pm 1.01$ |
| | 0.010 | 25 | 0.5 | $91.99 \pm 0.22$ | $2.52 \pm 0.64$ |
| | 0.020 | 25 | 0.5 | $90.05 \pm 0.37$ | $1.91 \pm 0.55$ |

## C  Additional Experimental Results

### C.1  The Performance under Different Datasets

Table 9 shows the detection results using two larger datasets, Tiny-ImageNet and Imagenet200 (please refer to Appendix C.2 for results on GTSRB). The ImageNet200 is a subset of ImageNet-1K. We train 10 models for each case, 60 models in total. BAN still performs well under the two larger datasets. In addition, BAN is also scalable in terms of time consumption. To conduct backdoor detection on ResNet18 models trained using CIFAR-10, Tiny-ImageNet, and ImageNet200, BAN takes around 55, 129, and 1,120 seconds, respectively. We do not apply other baselines (except for BTI-DBF) to larger datasets due to high computational requirements. For example, FeatureRE costs more than 2 hours for detection in one class of ImageNet200 (default settings), meaning FeatureRE needs more than 400 hours for a full detection on ImageNet200.

Table 9: The detection results using ResNet18 under different datasets.

| Dataset | Attack | BTI-DBF | | BTI-DBF* | | Ours | |
| | | Bd. | Acc. | Bd. | Acc. | Bd. | Acc. |
|---|---|---|---|---|---|---|---|
| | No Attack | 0 | 100% | 0 | 100% | 0 | 100% |
| Tiny-ImageNet | WaNet | 4 | 40% | 6 | 60% | 10 | 100% |
| | IAD | 9 | 90% | 8 | 80% | 10 | 100% |
| | No Attack | 3 | 70% | 0 | 100% | 0 | 100% |
| ImageNet200 | WaNet | 6 | 60% | 9 | 90% | 10 | 100% |
| | IAD | 10 | 100% | 10 | 100% | 10 | 100% |

### C.2  Dection on GTSRB

Table 10 shows the detection success rate for BAN on GTSRB. Notice that NC and BTI-DBF* perform worse against advanced attacks, while BAN precisely detects all backdoor models.

### C.3  Defense of Adaptive Attacks

Table 11 shows the fine-tuning results using our defense method against the two adap-blend and SSDT. The results show that fine-tuning with neuron noise can remove the backdoor with a slight decrease in clean accuracy.

### C.4  Comparsion with ANP

ANP [41] uses a similar neuron noise for pruning backdoor neurons. Their optimization objective includes neuron noise and neuron mask. The neuron noise is optimized to fool the network, while

Table 10: The detection results on GTSRB with ResNet18.

| Attack | NC Bd. | NC Acc. | BTI-DBF* Bd. | BTI-DBF* Acc. | Ours Bd. | Ours Acc. |
|---|---|---|---|---|---|---|
| BadNets | 20 | 100% | 20 | 100% | 20 | 100% |
| WaNet | 8 | 40% | 18 | 90% | 20 | 100% |
| IAD | 0 | 0% | 20 | 100% | 20 | 100% |
| Bpp | 13 | 65% | 14 | 70% | 20 | 100% |

Table 11: Fine-tuning for adaptive attacks.

| Attack | No defense BA | No defense ASR | Ours BA | Ours ASR |
|---|---|---|---|---|
| Adap-Patch | 94.20 | 99.75 | 90.45 | 10.24 |
| SSDT | 93.86 | 90.30 | 93.29 | 0.90 |

the mask controls a trade-off for clean cross-entropy loss. We compare our fine-tuning with ANP in Table 12. The hyperparameters of ANP are the default from the original paper.

Table 12: Comparison with ANP [41] on CIFAR-10 using ResNet18.

| Defense | BadNets BA | BadNets ASR | Blend BA | Blend ASR | WaNet BA | WaNet ASR | IAD BA | IAD ASR | Bpp BA | Bpp ASR |
|---|---|---|---|---|---|---|---|---|---|---|
| No defense | 93.37 | 99.41 | 94.60 | 100 | 93.57 | 99.37 | 92.83 | 97.10 | 94.29 | 99.93 |
| ANP | 93.16 | 2.13 | 94.17 | 97.24 | 93.06 | 13.84 | 92.50 | 0.44 | 93.80 | 4.77 |
| Ours | 92.26 | 2.04 | 92.61 | 1.04 | 92.18 | 0.87 | 92.36 | 1.47 | 92.82 | 2.16 |

## C.5   Different Poisoning Rates

BAN is effective against BadNets with different poisoning rates, as shown in Table 13. We use BadNets because it is relatively weak at a low poisoning rate (i.e., hard to be mitigated), while more advanced attacks may still be strong at a low poisoning rate.

Table 13: The performance of BAN fine-tuning under different poisoning rates of BadNets using CIFAR-10.

| Poisoning Rate | BA | ASR | pos. feature loss | neg. feature loss | Mitigated BA | Mitigated ASR |
|---|---|---|---|---|---|---|
| 0.01 | 93.48 | 98.69 | 0.38 | 0.17 | 92.07 | 2.73 |
| 0.05 | 93.37 | 99.41 | 0.37 | 0.35 | 92.06 | 1.97 |
| 0.10 | 90.98 | 100.00 | 0.35 | 2.06 | 90.29 | 2.17 |
| 0.15 | 90.32 | 100.00 | 0.39 | 2.23 | 90.16 | 1.71 |
| 0.20 | 89.34 | 100.00 | 0.44 | 2.43 | 90.39 | 1.01 |
| 0.25 | 88.09 | 100.00 | 0.56 | 2.81 | 89.55 | 1.54 |
| 0.30 | 86.09 | 100.00 | 0.62 | 3.13 | 88.83 | 1.08 |
| 0.40 | 82.39 | 100.00 | 0.67 | 3.51 | 88.75 | 1.67 |
| 0.50 | 77.83 | 99.97 | 0.84 | 4.27 | 86.87 | 3.56 |

## C.6   Against Backdoors on the MLP

We evaluate the defense performance on simple model architecture. In particular, a 4-layer MLP is trained with benign samples and with BadNets. Table 14 demonstrates that BAN is effective on MLP, where the "num. to target" refers to the number of samples (in 5000 validation samples) that are classified as the backdoor target after our detection. We also find that the positive feature loss (i.e., benign feature loss) is very close to the negative loss (i.e., potential backdoor feature loss), which indicates that the backdoor features are more challenging to decouple from benign ones.

## C.7   Discussion about Relationship between the Neuron Noise and Lipschitz Continuity

A small trigger that changes the output of a benign input into a malicious target label can be related to the high Lipschitz constant [45] and a neural network with high robustness tends to have a lower local Lipschitz constant. Moreover, a larger local Lipschitz constant implies steeper output around trigger-inserted points, leading to a smaller trigger effective radius making trigger inversion less

Table 14: The performance of BAN on the 4-layer MLP.

| MLP | BA | ASR | Pos. feature loss | Neg. feature loss | Num. to target | Mitigated BA | Mtigated ASR |
|---|---|---|---|---|---|---|---|
| Benign | 53.24 | - | 1.95 | 2.29 | 419 | - | - |
| BadNets | 47.04 | 100.00 | 2.03 | 2.34 | 3607 | 45.13 | 7.11 |

effective [49]. Thus, the concepts of adversarial activation noise and Lipschitz continuity are related, and the local Lipschitz constant can serve as an upper bound for the trigger's effective influence.

In addition, introducing theoretical tools like the Lipschitz constant for a backdoor defense may also be very tricky in practice because it needs approximation for implementation. For example, [45] evaluates the channel-wise Lipschitz constant by its upper bound but does not thoroughly discuss the relationship between the channel-wise Lipschitz constant and the network-wise Lipschitz constant, where the theorem of Lipschitz continuity really relies on. Recent work also mentions that empirically estimating the Lipschitz constant is hard from observed data, which usually leads to overestimation [5]. Methods relying on Lipschitz continuity may not require heavy computational load and are also related to our approach. Our method emphasizes more on fine-tuning with the guidance of neuron noise, rather than tuning the trained model.

## C.8 Choosing the $\lambda_1$

As discussed in the method section, the mask norm ($L_1$ norm) with lambda in Eq. (5) is to ensure that the optimization objective is decoupling between benign and backdoor features. Without the constraint of the mask norm in Eq. (5), the optimization objective will be simply increasing the mask norm unless there are extremely strong backdoor features. The $\lambda_1$ value selection is implemented by checking the value of the mask norm. Table 15 provides an example of $\lambda_1$ selection on CIFAR-10, where the maximal mask norm is 8192. It can be observed that the mask is almost full of ones when $\lambda_1$ is smaller than 0.7, and the negative feature loss (backdoor feature) is high, based on which we can pick a value of $\lambda_1$ greater than 0.7. Note that the selection is unaware of potential backdoors.

Table 15: Feature loss values with different $\lambda_1$

| $\lambda_1$ | Mask norm | Pos. feature loss | Neg. feature loss |
|---|---|---|---|
| 0.0 | 8188.62 | 0.27 | 2.30 |
| 0.1 | 8188.75 | 0.28 | 2.30 |
| 0.2 | 8184.30 | 0.27 | 2.30 |
| 0.3 | 8175.50 | 0.29 | 2.30 |
| 0.4 | 8152.40 | 0.25 | 2.30 |
| 0.5 | 8131.26 | 0.27 | 2.29 |
| 0.6 | 8055.07 | 0.21 | 2.27 |
| 0.7 | 7898.25 | 0.26 | 2.24 |
| 0.8 | 596.85 | 0.99 | 0.23 |
| 0.9 | 22.08 | 2.33 | 0.28 |

## C.9 Swin Transformer

Table 16 provides experimental results of a 12-layer Swin transformer on CIFAR-10. For BadNets and Blend, we train the backdoored network using Adam as the optimizer. For WaNet, IAD, and Bpp, we use the default setting. All backdoored models can be detected by BAN.

## C.10 Clean Label Attack

We provide a clean-label backdoor experiment following [35], where we take default hyperparameters. Table 17 demonstrates the detection performance of 20 models, where we randomly select one model from the 20 for the mitigation experiment. Experimental results demonstrate that BAN is also effective against the clean-label attack.

Table 16: Results on Swin Transformer.

| Attack | No defense | | FT | | BTI-DBF(U) | | BAN Fine-tune | |
|--------|------|-------|-------|--------|-------|--------|-------|-------|
| | BA | ASR | BA | ASR | BA | ASR | BA | ASR |
| BadNets | 86.7 | 97.43 | 82.23 | 44.07 | 85.21 | 99.99 | 83.92 | 2.91 |
| Blend | 85.46 | 100.00 | 80.13 | 100.00 | 83.9 | 100.00 | 79.18 | 26.20 |
| WaNet | 78.25 | 31.03 | 75.63 | 2.3 | 73.83 | 2.61 | 75.28 | 3.1 |
| IAD | 76.35 | 88.71 | 74.58 | 54.63 | 74.29 | 61.16 | 76.26 | 9.8 |
| Bpp | 79.19 | 63.74 | 75.35 | 12.99 | 74.66 | 11.04 | 74.25 | 6.91 |

Table 17: Clean label attack on CIFAR-10.

| Attack | BA | ASR | Detection Success Accuracy | Mitigated BA | Mitigated ASR |
|--------|-------|--------|----------------------------|--------------|---------------|
| LC [35] | 92.72 | 100.00 | 90.00 | 89.27 | 6.16 |

Table 18: Results on different target classes on CIFAR-10.

| Target | No defense | | BAN Fine-tune | |
|--------|------|--------|------|------|
| | BA | ASR | BA | ASR |
| 0 | 93.41 | 100.00 | 92.57 | 1.56 |
| 1 | 93.51 | 100.00 | 92.53 | 0.84 |
| 2 | 93.54 | 99.99 | 91.49 | 1.08 |
| 3 | 93.59 | 99.99 | 92.14 | 1.93 |
| 4 | 93.84 | 99.98 | 92.13 | 1.49 |
| 5 | 93.52 | 100.00 | 91.84 | 3.29 |
| 6 | 93.46 | 100.00 | 92.48 | 0.93 |
| 7 | 93.56 | 100.00 | 92.36 | 0.73 |
| 8 | 93.57 | 99.89 | 92.31 | 0.58 |
| 9 | 93.36 | 100.00 | 91.65 | 2.40 |

## C.11   Semantically Similar Classes

We provide an additional analysis on image form semantically similar classes. From ImageNet200, we select class n02096294 (Australian terrier) as the target class, since there are 19 kinds of terriers in our ImageNet200 datasets, such as n02094433 (Yorkshire terrier) and n02095889 (Sealyham terrier). We train 10 backdoor networks using BadNets with different target classes. Table 18 demonstrates that BAN is effective against the backdoor regardless of semantically similar classes.

