# OpenReview forum: "BAN: Detecting Backdoors Activated by Adversarial Neuron Noise"
_NeurIPS.cc/2024/Conference — NeurIPS 2024 poster_

### Official Review · Reviewer_cerE · 2024-06-14

**Soundness:** 3
**Presentation:** 4
**Contribution:** 3
**Rating:** 7
**Confidence:** 4

**Summary:**

The paper proposes a novel detection and defense method for backdoored models. The new method is motivated by the finding that existing state-of-the-art trigger-inversion methods, like BTI-DBF, rely on strong backdoor features, which might not always be present, such as for  BadNet-type triggers. As a solution, the paper proposes BAN (Backdoors activated by Adversarial neuron Noise), a novel detection method based on the findings that models with backdoors are more sensitive to adversarial noise than benign ones, allowing the identification of neurons responsible for the backdoor function. Practically, BAN computes neuron-specific noise inspired by adversarial attacks like PGD to maximize the model's classification loss on clean and unseen data. The method tries to decouple benign and backdoor features by optimizing a sparse neuron mask on the loss behavior of the model. Intuitively, a backdoored model tends to predict the target class label for clean inputs and the adversarial perturbed neurons, whereas a benign model shows fewer misclassifications under comparable neuron noise. Evaluated against common backdoor attacks and defense methods, BAN demonstrates improvements to existing approaches and robustness against adaptive attacks.

**Strengths:**

- The proposed defense method adds an interesting aspect to existing backdoor defense methods by improving the feature decoupling between benign and backdoor features. Whereas the decoupling optimization is conceptionally simple (which is not bad at all), it demonstrates strong results on the evaluation benchmarks and beats existing detection and defense methods markedly.
- The paper is well-written and easy to follow. Most parts of the proposed method is sufficiently placed in existing research, and the open research problems and the proposed solution are clearly presented and described. I enjoyed reading the paper.
- The evaluation is comprehensive and compares the proposed method against numerous attacks and defenses. It also includes the all-to-all setting, which is often ignored in literature. Evaluations of adaptive attacks underline the effectiveness and robustness of the approach.
- Empirical solutions and theoretical considerations sufficiently support all claims in the paper.

**Weaknesses:**

- The evaluation focuses on common (comparably shallow) CNN architectures. However, given that ViT also plays an increasing role in image classification, showing the effectiveness of the approach on ViT architectures in contrast to traditional CNNs would further support the efficiency claims of the method.
- The evaluation further focuses on dirty label attacks, i.e., attacks that change the label of a training example. However, the method's efficiency on clean-label attacks is not demonstrated. Including 1-2 clean label attacks in the evaluation would further strengthen the results.
- Some contributions of the paper should be stated more clearly and set apart from existing approaches in literature. For example, the feature decoupling process described in 3.3 is quite similar to the method by Xu et al. [1], and it is unclear to me (by only reading this paper) what exactly distinguishes the proposed method from the one in the existing literature.

Small remarks:
- Some captions are missing details. For example, the caption of Table 5 does not clearly state the investigated setting. Which dataset and model architecture are used here?
- The font in the figures could be increased. Some legends and texts are hard to read without zooming in.
- There is a typo in line 142: "experiemnts" -> experiments.
- Table 7 seems like a sensitivity analysis (measuring the sensitivity of the method to the hyperparameter selection) instead of an ablation study (impact of deactivating certain parts of the method).

[1] Xu et al. "Towards Reliable and Efficient Backdoor Trigger Inversion via Decoupling Benign Features". ICLR 2024

**Questions:**

- L303: "FeatureRE is designed for all-to-one attacks, so we use target label 0 here for FeatureRE." -> Does it make sense to include an approach designed for all-to-one attacks in an all-to-all setting? The comparison here with the other approaches seems misleading.
- Table 4: What are the BA (benign accuracy) and ASR (attack success rate) of models trained only on clean data without any poisoned data samples? It would be interesting to compare the defended backdoored models to a clean model. Since ImageNet (and its subsets) are complex datasets to learn, there will also be misclassifications that might, by chance, lead to the prediction of target class labels (even without any backdoor integration). Given ASR of the defended models, it would be interesting to see a baseline ASR (achieved only by random misclassifications) to assess if the remaining ASR after applying the defense methods is due to remaining backdoor behavior or simple random model behavior.
- The proposed method is based on the assumption that under neuron noise, a backdoored model tends to predict the target class label. In contrast, the predicted class labels for the benign model are rather equally distributed. This makes sense for datasets with clearly distinguishable classes, e.g., CIFAR10. Let us assume that there exist two classes in the dataset that are semantically very similar, e.g., two visually similar (but still different) dog breeds among other clearly separable classes (bug, plane, etc.). Let us further assume that the backdoor target is to change the label from the one dog class to the other one. If we now apply the neuron noise approach to a benign and a backdoored model, can we still assume that only the backdoored model will tend to predict the other dog class (the target class) under neuron noise? Or will the benign model under neuron noise behave similarly, since the other dog class is probably close in the feature space, and adding noise to the neuron activations could lead to predictions for the second dog class (which happens to be also the backdoor target)? Phrased differently, can we assume the proposed BAN method also works reliably on datasets with semantically similar classes, if the backdoor targets label changes from one class to the other?

**Limitations:**

Limitations are sufficiently discussed in Appx. A. Since the paper proposes a novel defense method for backdoor attacks, no negative societal impact is expected.

---

> ### Author Rebuttal · Authors · 2024-08-07
>
> Thank you for your valuable suggestions. We address your comments as follows.
>
>
> **Q1.** The evaluation only focuses on CNN architectures.
>
> **A1:** We provide ViT results on a 12-layer Swin transformer in the following table.
> For BadNets and Blend, we train the backdoored network using Adam as the optimizer.
> For WaNet, IAD, and Bpp, we use the default setting.
> All models in the following table are successfully detected as backdoored models.
> In addition, we notice that the attack performance of WaNet, IAD, and Bpp is not as good as BadNets and Blend.
> We conjecture this is because of training with SGD.
>
>
> | Attack | No def. (BA) | No def. (ASR) |FT (BA) | FT (ASR)| BTI-DBF (BA) | BTI-DBF (ASR) | Ours (BA) | Ours (ASR) |
> | ---- | ---- | ---- | ---- | ---- | ---- | ---- | ---- | ---- |
> | BadNets | 86.7 | 97.43 | 82.23 | 44.07 | 85.21 | 99.99 | 83.92 | 2.91 |
> | Blend | 85.46 | 100 | 80.13 | 100 | 83.9 | 100 | 79.18 | 26.20 |
> | WaNet | 78.25 | 31.03 | 75.63 | 2.3 | 73.83 | 2.61 | 75.28 | 3.1 |
> | IAD | 76.35 | 88.71 | 74.58 | 54.63 | 74.29 | 61.16 | 76.26 | 9.8 |
> | Bpp | 79.19 | 63.74 | 75.35 | 12.99 | 74.66 | 11.04 | 74.25 | 6.91 |
>
> **Q2.** The evaluation further focuses on dirty label attacks
>
> **A2:** Our submitted draft did not include clean-label attacks because it has a stricter threat model, which leads to weaker attacks.
> Here, we provide clean-label backdoor experiments to address the reviewer's concern using a well-known clean-label attack [A].
> We take default hyperparameters from the original paper.
> The following table demonstrates the detection and mitigation performance on 20 networks.
> We use one of the 20 networks for the mitigation experiment.
> Experimental results demonstrate that BAN is also effective against the clean-label attack.
>
>
> | Attack | BA | ASR | Detection Success Accuracy| Mitigation (BA) | Mitigation (ASR) |
> | ---- | ---- | ---- | ---- | ---- | ---- |
> |Clean Label[A] | 92.72 | 100 | 90\% | 89.27 | 6.16 |
>
>
>
> **Q3.** Does it make sense to include an approach (FeatureRE) designed for all-to-one attacks in an all-to-all setting?
>
> **A3:** The results of featureRE in Table 5 are to show that methods based on all-to-one settings may not work for all-to-all.
> Therefore, we believe it is not misleading since the featureRE relies on the assumption that backdoor attacks are always all-to-one. We will clarify this in the revised version.
>
>
> **Q4.** Some contributions of the paper should be stated more clearly and set apart from existing approaches in literature.
>
> **A4:** BTI-DBF [1] assumes the benign and backdoor features can be decoupled by a mask for benign features and (1-mask) for backdoor features.
> Then, they conduct trigger inversion based on the feature mask.
> However, we show in the following table that their mask will be a dense matrix full of ones.
> The feature's shape is $512\times4\times4$ (ResNet18), meaning the maximum mask norm is 8192.
> The reason is that the benign and backdoor features are not prominent to each other for weaker attacks, such as BadNets and Blend.
> In the BadNets (w/o norm), the negative feature loss is relatively larger because the (1-mask) is almost zero, and the loss is calculated by the feature $\times$ (1-mask).
> To solve this problem, we apply a $L_1$ maks norm to the feature mask decoupling.
> In addition, the decoupling performance can be further improved by adversarial neuron noise.
> We use the neuron noise to activate the backdoor such that the backdoor features are more prominent to the feature mask.
>
>
> |Attack| BA | ASR | L_1 mask norm | pos. feature loss | neg. feature loss|
> | ---- | ---- | ---- | ---- | ---- | ---- |
> |BadNets (w/ norm)| 93.47 | 99.71 | 2258.90 | 0.21 | 0.26 |
> |BadNets (w/o norm)| 93.47 | 99.71 | 8054.45 | 0.14 | 2.17 |
> |Blend (w/ norm) | 94.60 | 100 | 2084.62 | 0.15 | 0.20 |
> |Blend (w/o norm) | 94.60 | 100 | 8117.90 | 0.04 | 2.22 |
>
>
> **Q5.** Table 4: What are the BA (benign accuracy) and ASR (attack success rate) of models trained only on clean data without any poisoned data samples? It would be interesting to compare the defended backdoored models to a clean model.
>
> **A5:** We provide ASR on a benign model using CIFAR-10 and ResNet18 in the following table.
> The ASR of other attacks is omitted because attacks such as IAD train a generator for the backdoor trigger, which does not apply to a benign model.
> In the table, it is clear that ASR for the benign model is very low since the benign accuracy is high.
>
> |trigger type | BA | ASR |
> | ---- | ---- |  ---- |
> | BadNets | 94.77 | 0.56 |
> | Blend | 94.77 | 0.02 |
>
>
> **Q6.** can we assume the proposed BAN method also works reliably on datasets with semantically similar classes?
>
> **A6:** The classes with semantic similarities are actually widespread.
> The phenomenon makes classification results of these similar classes closely related. For example, Table 5 of IB-RAR [B] shows cats and dogs (CIFAR-10 classes) tend to be classified as each other under adversarial attacks.
> For ImageNet200, we use class n02096294 (Australian terrier) as the target class.
> There are 19 kinds of terriers in our ImageNet200 datasets,  such as n02094433 (Yorkshire terrier) and n02095889 (Sealyham terrier).
>
> To further support this observation, we train 10 backdoor networks using BadNets with different target classes.
> In the following table, BAN is contentiously effective against the backdoor regardless of semantically similar classes.
>
> | target | BA | ASR | mitigation (BA)| mitigation (ASR) |
> | ---- | ---- |  ---- | ---- |  ---- |
> | 0 | 93.41 | 100 | 92.57 | 1.56 |
> | 1 | 93.51 | 100 | 92.53 | 0.84 |
> | 2 | 93.54 | 99.99 | 91.49 | 1.08 |
> | 3 | 93.59 | 99.99 | 92.14 | 1.93 |
> | 4 | 93.84 | 99.98 | 92.13 | 1.49 |
> | 5 | 93.52 | 100 | 91.84 | 3.29 |
> | 6 | 93.46 | 100 | 92.48 | 0.93 |
> | 7 | 93.56 | 100 | 92.36 | 0.73 |
> | 8 | 93.57 | 99.89 | 92.31 | 0.58 |
> | 9 | 93.36 | 100 | 91.65 | 2.40 |
>
> [A] Label-consistent backdoor attacks.
>
> [B] IB-RAR: Information Bottleneck as Regularizer for Adversarial Robustness

---

> ### Comment · Reviewer_cerE · 2024-08-08
>
> I thank the authors for the detailed rebuttal and additional insights. All my questions and remarks were addressed. After reading all other reviews and author responses, I decided to increase my rating.

---

> > ### Author Response · Authors · 2024-08-08
> > **Thank you**
> >
> > Thank you for your quick reply and raising the score. We are happy that we have addressed your concerns

---

### Official Review · Reviewer_Eja2 · 2024-06-26

**Soundness:** 3
**Presentation:** 3
**Contribution:** 2
**Rating:** 6
**Confidence:** 4

**Summary:**

This paper provides an in-depth analysis of the SOTA trigger inversion-based backdoor defenses and finds that they suffer from high computational overhead and rely on prominent backdoor features. The authors tackle the challenges based on the previous findings on adversarial noise incorporating activation information and propose to improve the backdoor feature inversion for backdoor detection. The experiments show its efficiency and effectiveness compared to the SOTA methods.

**Strengths:**

1. The proposed method is intuitively reasonable and empirically effective. Both backdoor detection and defense are carefully considered.
2. The paper writing is clear and well-structured.
3. The experiments are comprehensive.

**Weaknesses:**

1. The motivations of high computational overhead and reliance on prominent backdoor features are not well-illustrated in the method section. Only the limitations of BTI-DBF are discussed.
2. The novelty is limited. The finding on neuron noise is previously proposed, and an additional mask regularizer based on BTI-DBF is a commonly used technique, which was used in NC as well. The idea of backdoor defense is similar to ANP, with the change in learning from the noise.

**Questions:**

None.

**Limitations:**

1. The reliance on prominent backdoor features is expected to be illustrated in detail, and the reason why the proposed method is able to overcome it with only a mask regularizer is unclear.  Some case visualizations are expected.
2.  The conclusion from Equation 7 that ignoring the second term is expected to be illustrated with evidence such as the separate loss values on it.

---

> ### Author Rebuttal · Authors · 2024-08-07
>
> Thank you for your valuable comments. We address your comments as follows.
>
> **Q1.** The motivations of high computational overhead and reliance on prominent backdoor features are not well-illustrated in the method section. Only the limitations of BTI-DBF are discussed.
>
> **A1:** High computational overhead:
> Most existing methods conduct trigger inversion for every class to detect potential backdoors (e.g., NC, FeatureRE, or Unicorn), so detection algorithms need to be executed multiple times, each using a different class as the optimization objective.
> Our detection does not rely on class-wise optimization and only executes once to detect the potential backdoor, which significantly reduces the time consumption.
> Similarly, BTI-DBF also only executes once for detection, but BTI-DBF needs to pre-train a U-Net generator to generate trigger images, which is computationally heavier.
>
> Prominent backdoor features:
> The reliance on prominent backdoor features is illustrated in Figure 3.
> We show that benign and backdoor features are very close for BadNets but prominent for advanced attacks, such as WaNet, IAD, and BPP. Please also refer to the analysis under Q3.
>
>
> **Q2.** The novelty is limited. The finding on neuron noise is previously proposed, and an additional mask regularizer based on BTI-DBF is a commonly used technique, which was used in NC as well. The idea of backdoor defense is similar to ANP, with the change in learning from the noise.
>
> **A2:** Our main finding is that prominent backdoor features may not be suitable for identifying input space backdoors.
> Based on the findings, we show the generalization shortcomings of previous approaches, and we fix current approaches by introducing the regularizer with the further help of neuron noise.
> We would argue that both our findings and defenses are novel because we recognized and then tried to solve common and previously unknown shortcomings from previous research.
> Technically, we introduced a novel feature space regularizer (Eq. (5)) combined with adversarial neuron noise, which strategically increases the feature difference and, more importantly, helps the defense generalization.
>
>
> **Q3.** The reliance on prominent backdoor features is expected to be illustrated in detail, and the reason why the proposed method is able to overcome it with only a mask regularizer is unclear. Some case visualizations are expected.
>
> **A3:** We will revise and improve the contents in Section 3.3 and Section 4.1, together with Figure 1, to better illustrate our findings on prominent backdoor features.
> Regarding BAN, the mask regularizer's utility is limited, and it is only effective when combined with the adversarial neuron noise.
>
> We also provide new experimental analysis in the table below, showing that previous decoupling methods cannot easily pick up backdoor features of weak attacks, such as BadNets, as their backdoor features are not prominent.
> For example, when detecting without the $L_1$ regularizer (i.e., w/o norm), the negative feature loss of BadNets is high with a very large $L_1$ mask norm, while the Bpp has an even higher negative loss with a much smaller mask norm.
> Note that for BadNets, the negative loss is high with a high mask norm where almost all features are included.
> It indicates that BadNet backdoor features are less prominent than Bpp features, making it more challenging to decouple BadNets features. We will provide a discussion in the revised draft.
>
> (The table is also presented in the global rebuttal.)
> A note on the table: The shape of the feature mask is $512\times4\times4$, which means the maximum of the mask $L_1$ norm is 8192.
>
> |Attack| BA | ASR | $L_1$ mask norm | pos. feature loss | neg. feature loss|
> | ---- | ---- | ---- | ---- | ---- | ---- |
> |BadNets (w/ norm)| 93.47 | 99.71 | 2258.90 | 0.21 | 0.26 |
> |BadNets (w/o norm)| 93.47 | 99.71 | 8054.45 | 0.14 | 2.17 |
> |Blend (w/ norm) | 94.60 | 100 | 2084.62 | 0.15 | 0.20 |
> |Blend (w/o norm) | 94.60 | 100 | 8117.90 | 0.04 | 2.22 |
> |WaNet (w/ norm)| 93.88 | 99.63 | 7400.97 | 0.06 | 2.34 |
> |WaNet (w/o norm)| 93.88 | 99.63 | 7702.56 | 0.05 | 2.39 |
> |IAD (w/ norm)| 93.82 | 99.64 | 7898.91 | 0.03 | 2.25 |
> |IAD (w/o norm)| 93.82 | 99.64 | 7895.16 | 0.03 | 2.25 |
> |Bpp (w/ norm)| 94.56 | 99.97 | 7147.68 | 0.09 | 2.80 |
> |Bpp (w/o norm)| 94.56 | 99.97 | 7260.31 | 0.09 | 2.78 |
>
>
> **Q4.** The conclusion from Equation 7 that ignoring the second term is expected to be illustrated with evidence such as the separate loss values on it.
>
> **A4:** The table above shows the $L_1$ mask norm and separated feature losses.
> When optimizing without the penalty of mask norm, for relatively weaker attacks (BadNet and Blend), the mask is very dense because the $L_1$ mask norm values are close to the upper bound (8192).
> The negative feature loss values are also large for BadNets and Blend, but this is because the negative feature mask, i.e., (1-mask), is close to zero.
> Almost no feature is used to get the negative feature loss, i.e., (1-mask)$\times$feature in Eq. (5) leads to the high loss value.
> In other words, the high negative feature loss (without the mask penalty) is not because of backdoor features on BadNets and Blend.
>
> With the penalty of mask norm, the optimization of the feature mask focuses more on finding backdoor features rather than only increasing the proportion of benign features.
> In the table, when optimizing with the penalty of mask norm, BadNets and Blend provide small negative feature loss values compared to advanced attacks (WaNet, IAD, and Bpp), even with a larger (1-mask).
> A small mask norm means the mask provides a larger proportion on the negative mask, i.e., (1-mask).
> That is to say, there are fewer backdoor features, or the backdoor features are less prominent.
> In conclusion, the feature decoupling using the mask is not effective when the backdoor features are not prominent.

---

> > ### Comment · Reviewer_Eja2 · 2024-08-08
> >
> > Thanks for the author's efforts in rebuttal. I am satisfied with the discussion on the prominence of features and my concerns are addressed. I will raise my score.

---

> > > ### Author Response · Authors · 2024-08-08
> > > **Thank you**
> > >
> > > Thank you for your quick reply and raising the score. We are happy that we have addressed your concerns

---

### Official Review · Reviewer_U3ov · 2024-07-08

**Soundness:** 3
**Presentation:** 3
**Contribution:** 3
**Rating:** 7
**Confidence:** 4

**Summary:**

This paper addresses the problem of efficient backdoor defense using the backdoor inversion approach. The authors leverage the past work “TOWARDS RELIABLE AND EFFICIENT BACKDOOR TRIGGER INVERSION VIA DECOUPLING BENIGN FEATURES” (BTI-DBF) which recovers a mask in the feature space to locate prominent backdoor features and decouples benign and backdoor features in the mask. The authors make contributions by improving the BTI-DBF method and by incorporating extra neuron activation information into their method called “Backdoors activated by Adversarial neuron Noise” or BAN.

**Strengths:**

The strengths of the paper lie in

(1)	studying the trigger inversion-based detection methods

(2)	optimizing the performance of the BTI-DBF method by adding a regularizer to the loss function

(3)	incorporating adversarial activation noise into the BAN method

**Weaknesses:**

The weaknesses of the paper lie in

(1)	missing explanations of the performance. For example, why is featureRE in Table 3 performing better than BAN? What would be the impact of a lambda value other than 0.5? Why is the Blend attack always better detected by BTI-DBF* than by BAN?

(2)	missing relationship between the adversarial activation noise approach and input/feature perturbation approach?  How is the adversarial activation noise approach different from assuming Lipschitz continuity on the estimated input-output function?

**Questions:**

The authors should answer the questions posed as weaknesses of the paper.

**Limitations:**

The limitations of the work lie in the accuracy and computational complexity of BAN. The authors describe the limitations in Appendix A and focus the limitations on the availability of clean (benign) samples and a slight decrease in the accuracy for benign inputs after fine-tuning.

The authors state, “We also exploit the neuron noise to further design a simple yet effective defense for removing the backdoor, such that we build a complete defense framework.” The statement “we build a complete defense framework” is an overstatement in the paper.

---

> ### Author Rebuttal · Authors · 2024-08-07
>
> Thank you for your appreciation of our work. We address your comments as
> follows.
>
> **Q1.** missing explanations of the performance. For example, why is featureRE in Table 3 performing better than BAN? What would be the impact of a lambda value other than 0.5? Why is the Blend attack always better detected by BTI-DBF* than by BAN?
>
> **A1:** We agree that featureRE provides slightly better benign accuracy but fails to mitigate the backdoor ASR for BadNets, Blend, and Bpp, while our method is consistently effective.
> Regarding the difference between featureRE and BAN, we observed that commonly (e.g., BTI-DBF or BAN), including featureRE, there is a trade-off between benign accuracy and backdoor deletion, where the backdoor deletion task harms benign accuracy.
>
> About the $\lambda$ values:
> We provided experiments with different values, as shown in Table 7 in the appendix (we tested 0.2, 0.5, 0.8, and 1).
> We saw that as the lambda value increased, the BA dropped.
>
> For the Blend attack, as shown in Table 3, the Blend backdoor is only better detected by BTI-DBF* on ResNet18, but our method performs better on VGG16 and DenseNet121.
>
>
> **Q2.** missing relationship between the adversarial activation noise approach and input/feature perturbation approach? How is the adversarial activation noise approach different from assuming Lipschitz continuity on the estimated input-output function?
>
> **A2:** Thank you for this intriguing question. From the literature, it is known that for a backdoor attack, if we have a small trigger that changes the output of a benign input into a malicious target label, this behavior can be related to the high Lipschitz constant [A] and a neural network with high robustness then tends to have a lower local Lipschitz constant.
> Moreover, a larger local Lipschitz constant implies steeper output around trigger-inserted points, leading to a smaller trigger effective radius and making trigger inversion less effective [B].
> Thus, the concepts of adversarial activation noise and Lipschitz continuity are related, and the local Lipschitz constant can serve as an upper bound for the trigger’s effective influence. We will include a discussion about this in the revised version of the document.
>
> In addition, introducing theoretical tools like the Lipschitz constant for backdoor defense may also be very tricky in practice because it needs approximation for implementation. For example,  [A] evaluates the channel-wise Lipschitz constant by its upper bound but does not thoroughly discuss the relationship between the channel-wise Lipschitz constant and the network-wise Lipschitz constant, where the theorem of Lipschitz continuity really relies on. In another recent paper [C], it is also mentioned that empirically estimating the Lipschitz constant is hard from observed data, which usually leads to overestimation. We admit that methods relying on Lipschitz continuity may not require heavy computational load and are also related to our approach. However, our method emphasizes more on fine-tuning with the guidance of neuron noise, rather than tuning the trained model.
>
>
>
> [A] Zheng, R., Tang, R., Li, J., Liu, L. Data-Free Backdoor Removal Based on Channel Lipschitzness. ECCV 2022.
>
> [B] Rui Zhu, Di Tang, Siyuan Tang, Zihao Wang, Guanhong Tao, Shiqing Ma, Xiaofeng Wang, Haixu Tang. Gradient Shaping: Enhancing Backdoor Attack Against Reverse Engineering. NDSS 2024.
>
> [C] Data-Driven Lipschitz Continuity: A Cost-Effective Approach to Improve Adversarial Robustness
>
> **Q3.** The authors state, “We also exploit the neuron noise to further design a simple yet effective defense for removing the backdoor, such that we build a complete defense framework.” The statement “we build a complete defense framework” is an overstatement in the paper.
>
> **A3:** We agree with the reviewer that this claim may not be accurate in some scenarios, and we rephrase the sentence as follows to avoid confusion: "We also leverage the neuron noise to design a simple yet effective defense that removes the backdoor."

---

> > ### Comment · Reviewer_U3ov · 2024-08-11
> >
> > I have read the rebuttal.  I do not have additional comments assuming that the authors are going to make changes and add the extra references according to their rebuttal.

---

> > > ### Author Response · Authors · 2024-08-11
> > > **Thank you**
> > >
> > > Thank you for your reply. We are happy that we have addressed your concerns. We will revise our work according to the rebuttal.

---

### Official Review · Reviewer_hJLG · 2024-07-09

**Soundness:** 3
**Presentation:** 2
**Contribution:** 3
**Rating:** 5
**Confidence:** 2

**Summary:**

This paper focuses on the backdoor defense task. The proposed detection method includes neuron noise, which leverages the differing robustness between regular and backdoored neurons, and a feature decoupling process using a mask. Additionally, a backdoor defense method is proposed, which achieves improved efficiency and overall performance.

**Strengths:**

The exhibited efficiency is impressive, and the motivation behind neuron noise is intriguing. The dimensions of conducted experiments are relatively thorough.

**Weaknesses:**

The neuron noise approach appears reasonable and interesting, but the feature decoupling with a mask is a little questionable.

Additionally, the study does not include enough baselines, such as fine pruning, which can obscure the distinction between backdoored and regular neurons.

**Questions:**

1, I might have a misunderstanding, but my key question is that the paper emphasizes that previous methods overly rely on prominent backdoor features. However, the Feature Decoupling with Mask at the neuron level seems to rely even more on these prominent backdoor features. Some previous works, like fine pruning [1], show that neurons might not necessarily be decoupled.

2, Additionally, in Section 4.4, the paper claims that "the reason is that the backdoor features of SSDT are close to benign features in the feature space. It is difficult for other methods to distinguish between backdoor and benign features created by SSDT." However, the Feature Decoupling with Mask approach may not be effective in this context.

3, There is a minor issue that the experimental improvement does not seem significant, especially in Table 3 and Table 4.

4, Is the detection performance related to the pattern of the trigger or the training strength of the backdoor?

5, I'm curious about when a model architecture becomes less redundant for a given dataset. For instance, when we train DenseNet121 on CIFAR-10, the initial features are likely to be sparse. On the other hand, training an MLP model on CIFAR-10 makes feature decoupling more challenging. I wonder if using smaller models makes feature decoupling less effective.

6 I also believe that selecting the lambda value in Equation (5) is quite challenging and tricky. This is because it is essential to maintain performance and ensure it doesn't degrade significantly after applying the mask (1-m).

I would like to change my score if the questions are well answered.

[1] Fine-Pruning: Defending Against Backdooring Attacks on Deep Neural Networks

**Limitations:**

The limitations have been discussed.

---

> ### Author Rebuttal · Authors · 2024-08-07
>
> Thank you for your valuable comments. We address your comments as follows.
>
> **Q1.**  I might have a misunderstanding, but ... be decoupled.
>
> **A1:**  Our defense aims to activate the backdoor by neuron noise such that the backdoor networks behave differently from benign networks. The feature mask we use in Eq. (5) is improved by the penalty of mask norm, and it does depend on the prominent backdoor. However, we apply the feature mask to activated backdoors (networks with neuron noise), which makes the decoupling easier. The existence of backdoored neurons is validated in [A,B,C,D]. As there are both backdoor and benign neurons in backdoored networks and only benign neurons in benign networks, it is evident that they will be different when applying noise to those neurons. Note that using neuron noise is insufficient for backdoor detection, as shown in Figure 2.
>
> **Q2.**  Additionally, in Section 4.4, the paper claims that ...  in this context
>
> **A2:** The feature decoupling methods, including other methods such as BTI-DBF and featureRE, may not always be effective.
> However, different from other methods, we apply the feature mask to a network with neuron noise. The noise activates the potential backdoor, which makes our decoupling easier.
>
> **Q3.** There is a minor issue that ... Table 3 and Table 4
>
> **A3:** We agree with the reviewer that performance improvement is sometimes not substantial. Our aim is to mitigate the generalization shortcomings of previous approaches. In our experiments, BAN is the only effective approach for all types of attacks.
>
> **Q4:** Is the detection performance ... backdoor?
>
> **A4:**  BAN is designed to be unaware of the trigger pattern, and we included backdoor attacks that are based on different patterns in experiments, including BadNets, Blend, IAD, WaNet, BppAttack, Adap-blend, and SSDT.
>
> Regarding the backdoor training strength, we provide experimental results with different poisoning rates of BadNets in the following table since the poisoning rate is closely related to backdoor strength. We use BadNets because it is relatively weak at a low poisoning rate, while more advanced attacks may still be strong at a low poisoning rate. All models in the table are successfully detected and mitigated. Our new results indicate that a stronger backdoor is easier to detect.
>
>
> |PR| BA | ASR | pos. feature loss | neg. feature loss|Mtigation(BA)|Mtigation(ASR)|
> | ---- | ---- | ---- | ---- | ---- | ---- | ---- |
> | 0.01 | 93.48 | 98.69 | 0.38 | 0.17 | 92.07 | 2.73 |
> | 0.05 | 93.37 | 99.41 | 0.37 | 0.35 | 92.06 | 1.97 |
> | 0.10 | 90.98 | 100 | 0.35 | 2.06 | 90.29 | 2.17 |
> | 0.15 | 90.32 | 100 | 0.39 | 2.23 | 90.16 | 1.71 |
> | 0.20 | 89.34 | 100 | 0.44 | 2.43 | 90.39 | 1.01 |
> | 0.25 | 88.09 | 100 | 0.56 | 2.81 | 89.55 | 1.54 |
> | 0.30 | 86.09 | 100 | 0.62 | 3.13 | 88.83 | 1.08 |
> | 0.40 | 82.39 | 100 | 0.67 | 3.51 | 88.75 | 1.67 |
> | 0.50 | 77.83 | 99.97 | 0.84 | 4.27 | 86.87 | 3.56 |
>
> **Q5:** I'm curious about ... makes feature decoupling less effective
>
> **A5:**  To validate this hypothesis, we trained a 4-layer MLP with the BadNets attack and another benign MLP for the CIFAR-10 dataset. In the table, the ``num. to target'' refers to the number of samples (in 5000 validation samples) that are classified as the backdoor target after our detection. After BAN detection, we find BadNets MLP classifies 3607 samples as the target, while for benign MLP, it is 419.  It means that BAN detects the backdoor.
>
> We also find that the positive feature loss (benign) is very close to the negative loss (potential backdoo). It indicates that the backdoor features are more challenging to decouple from benign ones, as the reviewer hypothesized. Note that the performance drop of MLP might also make features challenging to recognize.
>
> |MLP| BA | ASR | pos. feature loss | neg. feature loss| num. to target|Mtigation(BA)|Mtigation(ASR)|
> | ---- | ---- | ---- | ---- | ---- | ---- | ---- | ---- |
> |Benign| 53.24 | - | 1.95 | 2.29 | 419 | -| - |
> |BadNets| 47.04 | 100 |2.03 | 2.34 | 3607| 45.13 | 7.11|
>
> **Q6.** I also believe... applying the mask (1-m)
>
> **A6:** In practice, selecting the lambda value is easy. As discussed in the method section, the motivation for using the mask norm ($L_1$ norm) with lambda in Eq. (5) is to ensure the optimization objective is decoupling between benign and backdoor features.
> Without the constraint of the mask norm in Eq. (5), the optimization objective will be simply increasing the mask norm unless there are extremely strong backdoor features. Therefore, we choose a lambda value for Eq. (5) such that the mask norm does not change significantly while optimizing.
>
> This selection is easy because we only need to check the value of the mask norm. For example, the following table shows the $L_1$ mask norm and positive and negative feature losses. The feature size is $512\times 4\times 4$, which means the maximum of the mask norm is 8192 (every value in the mask is one). It is clear that the mask is almost full of ones when $\lambda_1$ is smaller than 0.7, and the negative feature loss (backdoor feature) is ignored. Therefore, in this case, we need the $\lambda_1$ values to be greater than 0.7. The selection of the lambda values is unaware of potential backdoors.
>
> | $\lambda_1$ | mask norm | Pos. feature loss | Neg. feature loss |
> | ---- | ---- |  ---- | ---- |
> | 0.0 | 8188.62 | 0.268| 2.30 |
> | 0.1 | 8188.75 | 0.28 | 2.30 |
> | 0.2 | 8184.30 | 0.27 | 2.30 |
> | 0.3 | 8175.50 | 0.29 | 2.30 |
> | 0.4 | 8152.40 | 0.25 | 2.30 |
> | 0.5 | 8131.26 | 0.27 | 2.29 |
> | 0.6 | 8055.07 | 0.21 | 2.27 |
> | 0.7 | 7898.25 | 0.26 | 2.24 |
> | 0.8 | 596.85 | 0.99 | 0.23 |
> | 0.9 | 22.08 | 2.33 | 0.28 |
>
>
>
>
> [A] ABS: Scanning neural networks for back-doors by artificial brain stimulation, CCS 2019
>
> [B] Backdoor Scanning for Deep Neural Networks through K-Arm Optimization, ICML 2021
>
> [C] Reconstructive neuron pruning for backdoor defense, ICML 2023
>
> [D] Pre-activation distributions expose backdoor neurons, NeurIPS 2022

---

> > ### Comment · Reviewer_hJLG · 2024-08-11
> >
> > I appreciate the authors' detailed rebuttal and the additional insights provided. Although most of my questions have been addressed, I still find the novelty somewhat unconvincing to warrant an increase in my rating. However, I agree with accepting the paper.

---

> > > ### Author Response · Authors · 2024-08-11
> > > **Thank you**
> > >
> > > We appreciate your careful consideration of our work, and we are happy you consider that we adequately addressed your comments. Thank you again for your support.

---

### Official Review · Reviewer_yW41 · 2024-07-12

**Soundness:** 3
**Presentation:** 3
**Contribution:** 3
**Rating:** 7
**Confidence:** 3

**Summary:**

The authors propose a novel technique for detecting backdoor attacks on neural networks by incorporating extra neuron activation information to reduce the overhead from prior backdoor feature inversion methods. The experimental results show a higher detection rate on the tested datasets when compared to the prior work.

**Strengths:**

The paper is well written and easy to follow. The authors address a significant topic with the widespread adoption of neural networks for a wide variety of tasks. The experimental results affirm the design choices made by the authors in Section 3. Overall, it is a well written paper that addresses a significant area.

**Weaknesses:**

1. A figure of the proposed method outlined in Sections 3.1 and 3.3 could be a helpful tool to visualize the proposed method.
2. The in figure text in Figures 2 and 3 are too small and hard to read.
3. Some tables use % and others don't when report accuracy, e.g. Table 2 and Table 3 with no % symbol for the BA columns.
4. I think a table or figure further emphasizing the proposed changes would greatly improve the strength of this paper.

**Questions:**

1. What is the impact of $\lambda_2$ on the fine-tuning loss found in Equation 8?

**Limitations:**

Yes, in appendix A.

---

> ### Author Rebuttal · Authors · 2024-08-07
>
> Thank you for your appreciation of our work. We address your comments as follows.
>
> **Q1.** A figure of the proposed method outlined in Sections 3.1 and 3.3 could be a helpful tool to visualize the proposed method.
>
> **A1:** We will add a figure that illustrates the design of our method.
>
> **Q2.** The in figure text in Figures 2 and 3 are too small and hard to read. Some tables use \% and others don't when report accuracy, e.g. Table 2 and Table 3 with no \% symbol for the BA columns.
>
> **A2:** We will increase the font size and revise the table coherence.
>
>
> **Q3.** I think a table or figure further emphasizing the proposed changes would greatly improve the strength of this paper.
>
> **A3:** We will revise the details of different methods and show the differences between our method and baselines.
>
> **Q4.** What is the impact of $\lambda_2$ on the fine-tuning loss found in Equation 8?
>
> **A4:** The hyperparameter $\lambda_2$ controls the trade-off between benign accuracy (BA) and attack success rate (ASR).
> Increasing the $\lambda_2$ value will decrease BA but provide better defense performance.
> Table 7 shows the results with different $\lambda_2$ values, ranging from 0.2 to 1, where $\lambda_2=0.5$ provides the best defense performance.

---

> > ### Comment · Reviewer_yW41 · 2024-08-07
> >
> > Thank you for the responses to all of my questions. I have read the rebuttal and do not have any further questions myself.

---

> > > ### Author Response · Authors · 2024-08-07
> > > **Thank you**
> > >
> > > Thank you for your quick reply. We are happy that we have addressed your concerns

---

### Author Rebuttal · Authors · 2024-08-07

Dear reviewers and ACs,

We thank you for evaluating and providing thorough feedback on our work. We are glad to see that most reviewers rated the paper positively, agreeing on the topic relevance and results provided in our work, such as "The authors address a significant topic", "The experimental results affirm the design choices made by the authors" (reviewer yW41), "The exhibited efficiency is impressive, and the motivation behind neuron noise is intriguing" (reviewer hJLG), "The proposed method is intuitively reasonable and empirically effective" (reviewer Eja2), and "Empirical solutions and theoretical considerations sufficiently support all claims in the paper" (reviewer cerE).

We made our best efforts to address the remaining concerns, and have written individual responses.
We are confident that the reviewer feedback and the incorporated additional experiments and discussions have even further strengthened our paper.

We include new experiments to illustrate the phenomenon that prominent backdoor features exist for advanced attacks (WaNet, IAD, and BPP) but may not be for weaker attacks (BadNets, Blend).
Experimental results demonstrate that previous decoupling methods cannot easily pick up backdoor features from weak attacks, such as BadNets, as their backdoor features may not be prominent.
For example, when detecting without the $L_1$ regularizer (i.e., w/o norm), the negative feature loss of BadNets is high with a very large $L_1$ mask norm, while the Bpp has an even higher negative loss with a much smaller mask norm.
The high negative loss of BadNets is actually from the sparse feature mask rather than backdoor features, i.e., there are too many zeros in (1-mask).
It indicates that BadNet backdoor features are less prominent than Bpp features, making it more challenging to decouple BadNets features. We will add a discussion in the revised draft.

A note on the table: The shape of the feature mask is $512\times4\times4$, which means the maximum of the mask $L_1$ norm is 8192.

|Attack| BA | ASR | $L_1$ mask norm | pos. feature loss | neg. feature loss|
| ---- | ---- | ---- | ---- | ---- | ---- |
|BadNets (w/ norm)| 93.47 | 99.71 | 2258.90 | 0.21 | 0.26 |
|BadNets (w/o norm)| 93.47 | 99.71 | 8054.45 | 0.14 | 2.17 |
|Blend (w/ norm) | 94.60 | 100 | 2084.62 | 0.15 | 0.20 |
|Blend (w/o norm) | 94.60 | 100 | 8117.90 | 0.04 | 2.22 |
|WaNet (w/ norm)| 93.88 | 99.63 | 7400.97 | 0.06 | 2.34 |
|WaNet (w/o norm)| 93.88 | 99.63 | 7702.56 | 0.05 | 2.39 |
|IAD (w/ norm)| 93.82 | 99.64 | 7898.91 | 0.03 | 2.25 |
|IAD (w/o norm)| 93.82 | 99.64 | 7895.16 | 0.03 | 2.25 |
|Bpp (w/ norm)| 94.56 | 99.97 | 7147.68 | 0.09 | 2.80 |
|Bpp (w/o norm)| 94.56 | 99.97 | 7260.31 | 0.09 | 2.78 |


Except for providing an additional prominent feature analysis, we also discuss and analyze different poisoning strengths, different architectures (i.e., less redundant MLP and popular Swin ViT), details on hyperparameter selections, and clean-label backdoors.
New experimental analyses and discussions can be found in the responses to each reviewer.

We are looking forward to hearing from you, and we remain at your disposal should you have any comments/suggestions.

Best regards,

Authors of BAN

---

### Decision · Program_Chairs · 2024-09-25

**Decision:**

Accept (poster)

**Comment:**

This work proposes a defense against backdoor attacks that improve trigger-detection-based defenses. The opinions about this paper are all positive (three accept, one weak accept, and one borderline accept). The reviewers had various concerns with this paper. However, the majority of them were solved by the rebuttal. The only relevant weakness that persisted after the rebuttal is that the novelty of this paper is incremental as it is a combination and improvement of previously proposed techniques. Nevertheless, the reviewer agrees that its effectiveness and efficiency surpass the competitors' approaches and can be accepted. The authors are recommended to add the clarifications and extra experiments made in the rebuttal to the paper.